

# Air quality modelling in the Berlin-Brandenburg region using WRF-Chem v3.7.1: sensitivity to resolution of model grid and input data

Friderike Kuik[1,2], Axel Lauer[3], Galina Churkina[1], Hugo A.C. Denier van der Gon[4], Daniel Fenner[5], Kathleen A. Mar[1], and Tim M. Butler[1]

[1]Institute for Advanced Sustainability Studies, Potsdam, Germany
[2]University of Potsdam, Faculty of Science, Potsdam, Germany
[3]Deutsches Zentrum für Luft- und Raumfahrt (DLR), Institut für Physik der Atmosphäre, Oberpfaffenhofen, Germany
[4]TNO, Netherlands Organization for Applied Scientific Research, Utrecht, The Netherlands
[5]Technische Universität Berlin, Faculty VI - Planning Building Environment, Institute of Ecology, Chair of Climatology, Berlin, Germany

*Correspondence to:* Friderike Kuik (friderike.kuik@iass-potsdam.de)

**Abstract.** Air pollution is the number one environmental cause of premature deaths in Europe. Despite extensive regulations, air pollution remains a challenge, especially in urban areas. For studying summertime air quality in the Berlin-Brandenburg region of Germany, the Weather Research and Forecasting Model with Chemistry (WRF-Chem) is set up and evaluated against meteorological and air quality observations from monitoring stations as well as from a field campaign conducted in 2014.

The objective is to assess which resolution and level of detail in the input data is needed for simulating urban background air pollutant concentrations and their spatial distribution in the Berlin-Brandenburg area. The model setup includes three nested domains with horizontal resolutions of 15km, 3km, and 1km and anthropogenic emissions from the TNO-MACC III inventory. We use RADM2 chemistry and the MADE/SORGAM aerosol scheme. Three sensitivity simulations are conducted updating input parameters to the single-layer urban canopy model based on structural data for Berlin, specifying land use classes on

a sub-grid scale (mosaic option) and downscaling the original emissions to a resolution of ca. 1km x 1km for Berlin based on proxy data including traffic density and population density. The results show that the model simulates meteorology well, though urban 2m temperature and urban wind speeds are biased high and nighttime mixing layer height is biased low in the base run. We show that the simulation of urban meteorology can be improved when specifying the input parameters to the urban model, and to a lesser extent when using the mosaic option. On average, ozone is simulated reasonably well,

but maximum daily eight hour mean concentrations are underestimated, which is consistent with the results from previous modelling studies using the RADM2 chemical mechanism. Particulate matter is underestimated, which is partly due to an underestimation of secondary organic aerosols. $NO_x$ (=NO+$NO_2$) concentrations are simulated reasonably well on average, but nighttime concentrations are overestimated due to the model's underestimation of the mixing layer height, and urban daytime concentrations are underestimated. The daytime underestimation is improved when using downscaled, and thus locally

higher emissions, suggesting that part of this bias is due to deficiencies in the emission input data and their resolution. The results further demonstrate that a horizontal resolution of 3km improves the results and spatial representativeness of the model





compared to a horizontal resolution of 15km. With the input data (land use classes, emissions) at the level of detail of the base run of this study we find that a horizontal resolution of 1km does not improve the results compared to a resolution of 3km. However, our results suggest that a 1km horizontal model resolution could enable a detailed simulation of local pollution patterns in the Berlin-Brandenburg region if the urban land use classes together with the respective input parameters to the urban canopy model are specified with a higher level of detail and if urban emissions of higher spatial resolution are used.

# 1 Introduction

Despite extensive regulations, air pollution in Europe remains a challenging issue: causing up to 400 000 premature deaths per year in Europe (EEA, 2015), air pollution is the number one environmental cause of premature deaths (OECD, 2012). Especially in urban areas air pollution is a problem, with 97%-98% of the urban European population (EU-28) exposed to ozone levels higher than 8-hour average concentrations of $100\,\mu g\,m^{-3}$, which the World Health Organisation (WHO) recommends not to be exceeded for the protection of human health, and ca. 90% of the urban European population (EU-28) exposed to $PM_{2.5}$ (particulate matter with a diameter smaller than 2.5 $\mu$m) levels higher than the WHO-recommended annual mean of 10 $\mu g\,m^{-3}$ in 2011-2013 (EEA, 2016). Similarly, annual and hourly $NO_2$ limit values are still exceeded, mainly at measurement site close to traffic. In 2013, the European limit value of $40\,\mu g\,m^{-3}$ was exceeded at 13% of all stations, all of them situated at traffic or urban sites (EEA, 2016). In Berlin, measured $NO_2$ annual means exceeded the European limit value of the annual mean at all but one measurement sites close to traffic in 2014 (Berlin Senate Department for Urban Development and the Environment, 2015a). In addition, current controversies on $NO_2$ emissions from cars have triggered additional discussions on $NO_2$ in urban areas.

Numerical modelling is an important tool for assessing air quality from global to local scales. Over the last decades, air quality models have been used to understand the processes leading to air pollution as well as to build a basis for policies defining measures to improve air quality. With increasing computing capacities, model resolution has been increasing, and different types of 3D regional chemistry transport models are able to resolve relevant processes down to a horizontal resolution of ca. 1km x 1km (Schaap et al., 2015). At these resolutions, the models can be used to study the atmospheric composition in the urban background.

As a basis for modelling work assessing air quality in the Berlin-Brandenburg area, this study evaluates a setup with the online-coupled numerical atmosphere-chemistry model WRF-Chem (chemistry version of the Weather Research and Forecasting model, Skamarock et al., 2008; Fast et al., 2006; Grell et al., 2005). In the setup presented here, WRF-Chem is coupled with a single-layer urban canopy model (Chen et al., 2011; Loridan et al., 2010). We evaluate the model setup with respect to its skill in simulating meteorological conditions and air pollutant concentrations, with a focus on $NO_x$ (=$NO+NO_2$), but also evaluating for particulate matter ($PM_{10}$, $PM_{2.5}$) and $O_3$. The skill in simulating air quality in an online-coupled model is, besides the choice of the chemical mechanism, influenced by the prescribed emissions, the model resolution and the skill in reproducing the observed meteorology. The latter depends on the model resolution, on input data, such as land use data, and on parametrizations of the sub-grid scale processes, such as effects of urban areas on meteorology. The objective of this





study is to address which resolution and level of detail in the input data, including land use, emissions and parameters characterizing the urban area, is needed for simulating urban background air pollutant concentrations and their spatial distribution in the Berlin-Brandenburg area. This is done by evaluating the model results of three nested model domains at 15km, 3km and 1km horizontal resolutions as well as three sensitivity simulations, including updating the representations of the urban area

within the urban canopy model, taking into account a sub-grid scale parametrization of the land use classes, and downscaling the original emission input data from a horizontal resolution of ca. 7km to ca. 1km. In light of the high computational costs of running the model at a 1km horizontal resolution, it is particularly helpful to find out under which conditions using this model resolution can lead to improved results compared to coarser resolutions. This can directly help the design of future air quality modelling studies over the Berlin-Brandenburg region and other European urban agglomerations of similar extent.

The WRF-Chem model has been applied and evaluated in different modelling studies over Europe. For example, Tucella et al. (2012) evaluate a European setup at a horizontal resolution of 30km x 30km. Brunner et al. (2015) and Im et al. (2015b, a) analyse the performance of several online-coupled models, set up for the Air Quality Model Evaluation International Initiative (AQMEII) phase 2. Among the simulations for a European domain are seven with different setups of WRF-Chem, performed with a horizontal resolution of 23km x 23km. Commonly reported biases of WRF-Chem in comparison to observations from

synoptic surface stations include an underestimation of daily maximum temperatures and an overestimation of wind speed (Tucella et al., 2012; Brunner et al., 2015). Furthermore, Brunner et al. (2015) conclude that the representation of other meteorological parameters relevant to air quality simulations, such as solar radiation at the surface, precipitation and planetary boundary layer height, is still challenging. WRF-Chem tends to underestimate ozone daily maxima over Europe (Tucella et al., 2012) with especially pronounced under-predictions of observed ozone values exceeding policy guidelines (Im et al., 2015b).

They attribute the deficiencies to the simulated meteorology, the chemical mechanism and the chemical boundary conditions. Mar et al. (2016) evaluated the performance of WRF-Chem to a European domain with respect to ozone, comparing different chemical mechanisms. They concluded that the simulated ozone concentration strongly depends on the choice of chemical mechanism, and that RADM2 leads to an underestimations of observed ozone concentrations. $PM_{10}$ is underestimated by WRF-Chem as compared to regional background observations (Im et al., 2015a). Tucella et al. (2012) also report an under-

estimation of $PM_{2.5}$. Both studies give various reasons for the mismatch in PM model results and observations, including an underestimation of secondary organic species by the aerosol mechanisms applied. Im et al. (2015a) report an overestimation of nighttime $NO_x$ in some models, including WRF-Chem, which they attribute both to a general underestimation of $NO_2$ during low-$NO_x$ conditions and to problems in simulating nighttime vertical mixing. They report that $NO_2$ is underestimated by most models.

WRF-Chem has also been applied at high spatial resolutions over urban areas, for example Mexico City (Tie et al., 2007, 2010), Los Angeles (Chen et al., 2013), Santiago (Mena-Carrasco et al., 2012), the Yangtze River Delta (Liao et al., 2014) and Stuttgart (Fallmann et al., 2016). Tie et al. (2007, 2010) have explicitly assessed how the model resolution impacts the simulated ozone and ozone precursors in Mexico City and concluded that a resolution of 24km is not suitable for simulating concentrations of CO, $NO_x$ and $O_3$ in the city centre. They suggest a ratio of city size to model resolution of 6:1 and conclude

that a horizontal resolution of about 6km is the best balance between model performance and computational time when sim-





ulating ozone and precursors in Mexico City. Furthermore, they conclude that the model results for ozone are more sensitive to the model resolution than to the resolution of the emission input data. Other studies have shown that increasing the model resolution does not necessary lead to an improvement in model results, but that it can be beneficial for amplifying the urban signal (e.g. Schaap et al., 2015, and references therein). They emphasize that it is only useful to go to model resolutions finer

than 20km if model input data, such as land use data and emission data, is also available at similarly high resolutions. Fallmann et al. (2016) have combined WRF-Chem with RADM2 chemistry and MADE/SORGAM aerosols with a multi-layer urban canopy model for the area of Stuttgart, studying effects of urban heat island mitigation measures on air quality. One of their findings from the model evaluation is an underestimation of daytime $NO_2$ by up to 60%, while $O_3$ is slightly overestimated during the day.

In the Berlin-Brandenburg region, there have been regional model simulations of particulate matter with an offline chemistry transport model (Beekmann et al., 2007), along with a measurement campaign focusing on particulate matter in 2001/02. Other modelling studies in this region focused on meterology: Schubert and Grossman-Clarke (2013) assessed the impact of different measures on extreme heat events in Berlin. Trusilova et al. (2016) tested different urban parametrizations in the COSMO-CLM model and their impact on air temperature. Jänicke et al. (2016) used the WRF model to dynamically downscale global

atmospheric reanalysis data over Berlin to a resolution of 2kmx2km, testing combinations of different planetary boundary layer schemes and urban canopy models. They conclude that simulated urban-rural as well as intra-urban differences in 2m air temperature are underestimated and that the more complex urban canopy models did not outperform the simple slab/bulk approach.

  To our knowledge there are no published studies for the Berlin-Brandenburg region simulating chemistry and aerosols with

20 an online-coupled regional chemistry transport model. Furthermore, only few of the above-mentioned studies included an assessment of urban $NO_x$ concentrations. In light of the recent exceedences of $NO_2$ in European urban areas, including Berlin, this study can contribute to filling this gap and serve as a basis for future modelling studies addressing $NO_x$ in European urban areas.

## 2 Model setup

### 2.1 Model description, chemistry and physics schemes

For this study, we use the Weather Research and Forecasting model (WRF) version 3.7.1 (Skamarock et al., 2008), with chemistry and aerosols (WRF-Chem, Grell et al., 2005; Fast et al., 2006). We use three one-way nested model domains centred around Berlin, at horizontal resolutions of 15kmx15km, 3kmx3km and 1kmx1km (Fig. 1). The model top is at 50 hPa, using 35 vertical levels. The setup includes the RADM2 chemical mechanism with the Kinetic PreProcessor (KPP) and the

30 MADE/SORGAM aerosol scheme. We give the priority to using the KPP solver instead of the QSSA (quasi steady-state approximation) solver, because Forkel et al. (2015) found that the latter underestimates nighttime ozone titration for areas with high NO emissions. However, this option does not allow us to include the full aqueous phase chemistry, including aerosol-cloud interactions and wet scavenging, and might thus reduce the model skill in simulating aerosols formed through aqueous



phase reactions as reported in Tucella et al. (2012). All settings, including the physics schemes used in this study, are listed in table 1, and the namelist can be found in the supplementary material. We use the European Centre for Medium-Range Forecast (ECMWF) Interim reanalysis (ERA-Interim, Dee et al., 2011) with a horizontal resolution of 0.75°x0.75°, temporal resolution of 6h and 38 vertical levels as meteorological initial and lateral boundary conditions. This also includes the sea surface

temperature, which is updated 6-hourly. The data is interpolated to the model grid using the standard WRF preprocessing system (WPS). Chemical boundary conditions for trace gases and particulate matter are created from simulations with the global chemistry transport Model for Ozone and Related chemical Tracers (MOZART-4/GEOS-5, Emmons et al., 2010).

## 2.2   Land use specification

An analysis of the USGS land use data commonly used in WRF showed that the land cover of the region around Berlin is

not represented well. Therefore we implemented the CORINE dataset (EEA, 2014) to replace the USGS dataset. The original CORINE dataset includes 50 land use classes. The land use classes at the spatial resolution of 250 m are remapped to 33 USGS land use classes read by WRF, following suggestions of Pineda et al. (2004) (see also Table S1). Additionally we distinguish between inland water bodies (USGS class 28) and other water bodies (USGS class 16). We map the urban land use classes in CORINE to three urban classes used in WRF-Chem, including commercial/industry/transport (USGS class 33), high (USGS

class 32) and low (USGS class 31) intensity residential (Tewari et al., 2008), which can be characterized as follows: low intensity residential (31) includes areas with a mixture of constructed materials and vegetation. Constructed materials account for 30-80% of the cover and vegetation may account for 20 to 70% of the cover. These areas most commonly include single-family housing units, and population densities are lower than in high intensity residential areas. High intensity residential (32) includes highly developed areas with a high population density. Examples include apartment complexes and row houses. Vegetation ac-

counts for less than 20% of the area and constructed materials account for 80 to 100%. Commercial/industrial/transportation (33) includes infrastructure (e.g., roads, railroads, etc.) and all highly developed areas not classified as high intensity residential.

We implement the new land use categories as decribed in Tewari et al. (2008) (Fig. 2). In addition, we adjust the initialization of the dry deposition of gaseous species to account for these new land use categories, as described in Fallmann et al. (2016). For the base run we use the bulk approach of the land surface scheme, assigning the most abundant land use class within a

model grid cell to the whole grid cell. In a sensitivity simulation we test the mosaic approach (Li et al., 2013), allowing us to account for a heterogeneous land use classification within one model grid cell. Up to eight different land use types within one model grid cell are considered in our setup.

## 2.3   Urban parameters

We use the single-layer urban canopy model (Kusaka et al., 2001; Kusaka and Kimura, 2004) to account for the modified

dynamics by cities, especially Berlin and Potsdam. The urban model takes into account energy and momentum exchange between urban areas (roofs, walls, streets) and the atmosphere and is coupled to the Noah land surface model. Surface fluxes (heat, moisture) and temperature are calculated as a combination of fluxes from urban and vegetated surfaces, coupled via the urban fraction assigned to the land use type of the grid cell (Chen et al., 2004).





In our base simulation, we use the default input parameters. For a sensitivitiy simulation (Sect. 2.5), we calculate some of the urban input parameters to the model for Berlin (Table 2), which in previous studies have been found to be important. Geometric parameters include roof level building height, standard deviation of the roof height, roof width and road width. The calculations are based on detailed maps provided by the Senate Department for Urban Development and the Environment of Berlin. From the original data containing information on the location and number of floors of each house, the mean building height and the standard deviation of the building height is calculated assuming an average height of 3 m per floor, and the mean building length is calculated with the software QGIS. We combine this data with the CORINE land use data for Berlin mapped to the USGS classes (section 2.2), averaging these parameters over the parts of the city characterized by the same urban class. The maps further provide the location of individual road segments, which we use to calculate the total area covered by roads in Berlin. We combine this with the total length of all roads in Berlin (Berlin Senate Department for Urban Development and the Environment, 2011b) to obtain the average road width, which we assigne to all three urban land use categories. We further update the urban fraction, using a spatially more detailed classification of the land use types and the fraction of impervious surface of each area, provided by the Senate Department for Urban Development and the Environment of Berlin. Following Schubert and Grossman-Clarke (2013), we assume the urban fraction of a grid cell to be equal to the fraction of impervious surface. We then define the mean of impervious surface area, weighted by the area of the respective surface within each land use class as the updated urban fraction of the respective class. Following Fallmann et al. (2016) we use the values for thermal conductivity, heat capacity, emissivity and albedo of roofs, walls and streets specified in Salamanca et al. (2012).

## 2.4 Emissions

For the base run, anthropogenic emissions of CO, $NO_x$, $SO_2$, NMVOCs, $PM_{10}$, $PM_{2.5}$ and $NH_3$ are taken from the TNO-MACC III inventory, with a horizontal resolution of 0.125°x0.0625°. The inventory is based on nationally reported emissions for specific sectors, distributed spatially based on proxy data. In comparison to version II of the inventory (Kuenen et al., 2014), version III includes an improved distribution of emissions especially around cities. Seasonal, weekly and diurnal emission profiles for Germany are applied to the aggregated emissions. This, as well as the speciation of the different non-methane (NM) VOCs is described in Mar et al. (2016) and von Schneidemesser et al. (2016). Mar et al. (2016) found that distributing emissions vertically did not strongly impact the model results near the surface. This, and the low stack height of point sources within Berlin, is why in this study all emissions are released into the first model layer. As much of the $NO_x$ emitted within Berlin is emitted from diesel vehicles (off-road and on-road), which studies have shown to be composed of high proportions of $NO_2$ (e.g. Alvarez et al., 2008), $NO_x$ is emitted as 70% NO and 30% $NO_2$ (by mole). The latest available emission dataset is for 2011, which is used in the 2014 simulations. Dust, sea salt and biogenic emissions are calculated online, the latter using the Model of Emissions of Gases and Aerosols from Nature (MEGAN v2, Guenther et al., 2006).

We perform a sensitivity simulation for testing the model sensitivity to the spatial resolution of the emission input data (Sect. 2.5). As input to this sensitivity simulation we downscale the anthropogenic emissions within Berlin onto a grid of 1/7 of the original resolution based on two proxy datasets, including traffic densities and population (Berlin Senate Department for Urban Development and the Environment, 2011a, b). Traffic densities are used to downscale all emissions from road transport





and population data is used to downscale emissions from industry, residential combustion and product use. Point sources are included in the grid cell within which the point source is located. All emissions from the energy industry within Berlin are point sources, and point source emissions from other industry sectors amount to ca. 55% of the total emissions within Berlin for CO, 9-17% for particulate matter and up to 1% for other gases. Agricultural emissions within the city boundaries of Berlin

are close to zero, which is why these are used at the original resolution.

## 2.5 Model simulations

Simulations are done for summer 2014 (May 31 - August 28). We chose to simulate the summer of 2014, as this corresponds to the time period of the BAERLIN measurement campaign (e.g. Bonn et al., 2016). While mean observed temperatures in June and August showed little deviations from the observed 30-year mean (1961-1990) with mean temperatures of 17.0° (June) and

10 17.2°, the July mean temperature of 21.3° was 3.4°C higher than the 30-year mean. Precipitation was 12% and 13% lower than the 30-year mean in June (62.5mm) and July (60.2mm), respectively and it was 48% lower than the 30-year mean in August, with 33.8mm (Berlin Senate Department for Urban Development and the Environment, 2014a, b, c).

For the analysis, the first day of all simulations is discarded as spin-up. A base run with the settings described above is done in order to evaluate the model performance in simulating observed meteorology and atmospheric composition. In addition,

sensitivity simulations done for this study are the following, with the changes applied to all three model domains of horizontal resolutions of 15km, 3km and 1km:

- ○ S1_ urb: updated representation of the urban characteristics of Berlin (see Sect. 2.3 and Table 2),

- ○ S2_ mos: consideration of the heterogeneity of the land use categories within one model grid cell (mosaic approach, see Sect. 2.2),

- ○ S3_ emi: using emissions downscaled to ca. 1kmx1km (see Sect. 2.4).

The purpose of the sensitivity simulations is to assess which resolution and level of detail in the input data, including land use (S2_mos), emissions (S3_emi) and parameters characterizing the urban area (S1_urb), is needed for simulating urban background air pollutant concentrations and their spatial distribution in the Berlin-Brandenburg area, particularly focusing on $NO_x$. We particularly ask whether a horizontal model resolution of 1km, together with the above-listed specifications of the

25 input data, leads to model results that differ from those obtained with a horizontal resolution of 3km.

## 3 Observational data description and model evaluation procedure

### 3.1 Data description

In the following, we list the data and data sources that we use for evaluating the present WRF-Chem setup for Berlin and surroundings. Table 3 gives an overview over all observational data and measurement stations in Berlin and surroundings used

in this study.



### 3.1.1 DWD stations

We use observations from the German Weather Service (DWD) for the variables 2m temperature, 10m wind speed and direction and precipitation from stations within Berlin and Potsdam for 2014. A second quality control, as described in Kaspar et al. (2013), has been applied to the data. Additionally, we obtained mixing layer heights calculated from radiosonde observations

directly from the DWD at the Lindenberg station southeast of Berlin, as described in Beyrich and Leps (2012). In addition, we use data of the specific humidity from the Global Weather Observation dataset provided by the British Atmospheric Data Center (BADC) for the same stations.

### 3.1.2 TU stations

The Chair of Climatology of Technische Universität Berlin (TU) runs an urban climate observation network (Fenner et al.,

2014), from which we use observations of 2m air temperature to complement observations from DWD stations. We include this additional data source, as many of the TU stations are situated in urban built-up areas (see Table 3). Sensors (Campbell CS215, accuracy $\pm 0.4$ K in the range $5 - 40°$C) were calibrated in a climate chamber and measurement data at 1-min resolution were quality checked by firstly identifying unrealistic measurement values (i.e. 3K below or above long-term minima and maxima per month at the sites Berlin-Tempelhof, period 1948-2014 and Potsdam, period 1900-2014) and secondly by visual inspection.

Afterwards, the data was aggregated to hourly mean values.

### 3.1.3 GRUAN network

The Global Climate Observing System Upper-Air Network (GRUAN) hosts radiosonde observations at high vertical resolution, of which we use observations of temperature in Lindenberg (Sommer et al., 2012) to compare them to the modeled profiles. The data used for this study is quality-checked, processed and bias-corrected as described in Sommer et al. (2012); Dirksen

et al. (2014).

### 3.1.4 UBA database and BLUME network

Legally required air quality observations in Germany are reported to the Federal Environment Agency (UBA). We use observations of $PM_{10}$, $PM_{2.5}$, $NO_2$, NO and $O_3$ for 2014 reported to UBA. The data is collected from measurement networks operated by the Federal States. In Berlin, the official measurement network is the BLUME network (Berliner Luftgüte-Messnetz), oper-

25 ated by the Senate Department for Urban Development and the Environment of Berlin. In addition to the data reported to the UBA database, we use $PM_{10}$ concentrations measured at three stations in Berlin and the 2m temperature measured at the urban built-up station Nansenstraße from the BLUME network.

### 3.1.5 BAERLIN2014

The BAERLIN2014 („Berlin Air quality and Ecosytem Research: Local and long-range Impact of anthropogenic and Natural

hydrocarbons 2014") campaign took place in Berlin in summer 2014 and is described in detail in Bonn et al. (2016) and von



Schneidemesser et al. (manuscript in preparation). For the present study, we use observations of $PM_{2.5}$ calculated from particle number concentrations collected near the Nansenstraße station of the BLUME network and observations of the mixing layer height collected at Nansenstraße with a ceilometer. In addition, filter samples taken at Nansenstraße were analyzed for the composition of $PM_{10}$ (von Schneidemesser et al., manuscript in preparation), which we use to compare to simulated aerosols.

## 3.2 Model evaluation procedure

In order to assess the model's skill in simulating observed meteorology, we compare the modeled (coarse domain) weather types with weather types calculated from the ERA-Interim reanalysis data for Berlin (Sect. 4.1). The weather types are based on indices calculated to classify circulation patterns and are further described in Otero et al. (2016). We then focus on evaluating the modelled meteorology including 2m temperature (T2), 10m wind speed and direction (WS10 and WD10), the atmospheric structure via comparing temperature profiles and mixing layer height (MLH) as well as 2m specific humidity (Q2) and precipitation. While T2, WS10, WD10 and atmospheric vertical structure are important parameters for simulating atmospheric chemistry and aerosols, Q2 and precipitation will not have an impact on our results, as our setup does not include aqueous phase chemistry or wet scavenging. However, we include Q2 and precipitation to complete the picture of the evaluation of simulated meteorology as well as to give an indication for future studies based on this setup. Finally, we evaluate the model performance for the main air pollutants including surface $O_3$, $NO_x$ and PM, with a main focus on $NO_x$. We evaluate the model results from all three domains with horizontal resolutions of 15km, 3km and 1km, which we also refer to as d01, d02 and d03.

### 3.2.1 Comparison with surface station data

The evaluation of surface parameters is based on statistical metrics including the Pearson correlation coefficient (r), the mean bias (MB) and the normalized mean bias (NMB). The metrics are defined as follows, with n the number of model - observation pairs, M the modeled values, O the observations and $\sigma$ the standard deviation of modeled or observed values:

$$r = \frac{1}{(n-1)} \sum_{i=1}^{n} \left( \frac{M_i - \overline{M}}{\sigma_M} \right) \left( \frac{O_i - \overline{O}}{\sigma_O} \right)$$

$$MB = \frac{1}{n} \sum_{i=1}^{n} M_i - O_i$$

$$NMB = \frac{\sum_{i=1}^{n} M_i - O_i}{\sum_{i=1}^{N} O_i}$$

For the meteorological parameters, the metrics are calculated from instantaneous hourly modeled values and hourly averages of the observations. Wind speed is considered as a scalar and no metrics are calculated for wind direction. For $O_3$, $NO_x$ and PM we calculate them from daily averages. The NMB was only calculated for air pollutants and the mixing layer height. For ozone, we also consider the maximum daily eight hour mean (MDA8) concentrations, a metric used in the European Union's Air Quality Directive.

As an additional means of assessing the model performance we look at conditional quantile plots (Carslaw and Ropkins, 2012) for some species. The conditional quantile plot displays the model results, split into evenly spaced bins, in comparison



to observations temporally matching the values in the model result bins. Thus, it gives additional insight into how well the modeled values agree with the observations, e.g. on the range of modeled and observed values.

For the comparison between model and observations, we classify the stations in terms of their surroundings, distinguishing between urban built-up, urban green and rural for the meteorology observations, and between urban background, suburban
background and rural for air quality observations, excluding those from traffic stations.

### 3.2.2  Evaluation of the atmospheric structure

The mean modeled temperature profiles are compared to observations from radiosondes as follows: as the observed temperatures have a much higher spatial resolution than the model, we select a subset of the observations for comparison with the model. For every modeled temperature profile at 00:00, 06:00, 12:00 and 18:00 UTC, we select the observations closest to
the modeled geopotential height of each model level. The time-averaging of modeled geopotential heights is done as follows: we divide the values into vertical bins corresponding to the 5th, 10th, 15th, ... , 95th percentiles of the modeled geopotential height, and average the temperature as well as the geopotential height over each bin for both model and observations and over each day of the modeled period.

The modeled MLH is compared to observations in two different ways: firstly, using the planetary boundary layer height
directly diagnosed by WRF-Chem, and secondly, by calculating the MLH from the simulated profiles of temperature, wind speed and humidity, defining the mixing layer height as the height where the Richardson number is 0.2, following Beyrich and Leps (2012). This corresponds to the method the MLH is derived from radiosonde observations at Lindenberg.

## 4  Model evaluation results: base run

### 4.1  Meteorology

Generally, the modeled weather types (see Sect. 3.2) are consistent with those derived from the reanalysis (Fig. 3). Periods in which WRF-Chem weather types disagree with ERA-Interim weather types never exceed two subsequent days and the frequency of WRF-Chem weather types agrees similarly well with ERA-Interim weather types.

The temporal correlation of modeled hourly 2m temperature with observations is between 0.88 and 0.91 at all stations in and around Berlin and all model domains (Tables 4 and S3 in the supplementary material), which shows that the model
represents the observed temperature variability well. This is supported by the analysis of the conditional quantiles (Fig. 4), which show that the modeled temperatures match the observations well for a wide range of values. The model is generally biased positively with up to +1.6° C, though the bias at most stations is smaller than +1° C (Tables 4 and S3). In absolute terms, this is within the same range, but never larger than the biases that Trusilova et al. (2016) and Schubert and Grossman-Clarke (2013) found using COSMO-CLM in combination with different urban canopy models for Berlin. Besides, the absolute
mean biases are comparable to those reported by (Jänicke et al., 2016), who mainly found negative biases in near-surface air



temperature applying WRF 3.6.1 for Berlin and surroundings, testing two planetary boundary layer schemes and three urban canopy models.

The histogram in the conditional quantile plot and the extent of the blue line marking the „perfect model" show that WRF-Chem does not reproduce the highest observed temperatures. This suggests that the model might have difficulties in simulating
pronounced heat wave periods. However, comparing the modeled daily maximum temperatures to the observed daily maximum temperatures (Tables 5 and S4) shows that the bias of the daily maximum temperatures is of a similar magnitude as the mean bias, with one difference: while the bias of maximum temperatures modeled with 3km and 1km resolutions is mainly positive, the bias of the maximum temperatures modeled with a 1km resolution is negative. In absolute terms, the bias of the daily maximum temperatures is smallest for results obtained with a 1km resolution, though they only differ very little from the
results obtained with a 3km resolution.

We find two important relationships with respect to model resolution: Firstly, the model simulates higher temperatures in the model domain of which the model grid cell land use type is urban (stations Kaniswall, Dahlemer Feld, Marzahn, Schönefeld). Secondly, while the modeled temperatures generally differ between the 15km and 3km resolution even if the land use type of both grid cells, in which the station is located, is the same, the June-July-August (JJA) mean modeled temperature only
changes by more than 0.1°C between the 3km and 1km resolution if the land use type changes (stations Bamberger Straße, Nansenstraße, Schönefeld). This indicates that switching from a horizontal resolution of 15km to 3km might improve the spatial distribution of modeled temperatures, while switching from a horizontal resolution from 3km to 1km has only a very little effect on improving the model's skill in simulating the observed temperature, but might be more beneficial if the land use input data is specified with a higher level of accuracy.

The comparison of simulated with observed temperature profiles (Fig. 5) shows that the model reproduces the observed temperature profile well at all times, but that the modeled temperature profile at 12:00 UTC is shifted to higher temperatures by ca. 1°C. The result is similar for all model resolutions (the profiles for the 15km and 3km resolutions can be found in the supplementary material in Fig. S1 and S2). In order to further evaluate how the present WRF-Chem setup simulates the observed vertical structure, we compare the simulated mixing layer height derived from simulated profiles of temperature,
wind speed and humidity (in the following also referred to as MLH-calc) to the mixing layer height derived from radiosonde observations at Lindenberg as described in Beyrich and Leps (2012) (Fig. 6). The results show that the model simulates the observed diurnal cycle of the MLH as well as the magnitude of the observed MLH at Lindenberg reasonably well: the bias of the daily mean MLH ranges between +87m (13%) and +113m (16%), depending on model resolution, and the biases of the daily maximum and daily minimum are between +268m (19%) and +347m (25%) and between +26m (14%) and +48m
(26%), respectively (Table 6). There is no consistent trend with increasing model resolution. It is important to note that these results refer to the MLH that we calculated from simulated profiles of temperature, wind speed and humidity. However, the MLH diagnosed by the model, in the following also referred to as MLH-YSU, underestimates the observations especially at nighttime (Fig. 6), with a bias of the daily minimum MLH between -99m (-53%) and -113m (-60%), or a MLH lower than the calculated one between -128% and -214%. Differences between the different ways of deriving the MLH for daily
maximum values are less pronounced, ranging between 24m (1%) and 73m (4%). This leads to the conclusion that the model





generally simulates the atmospheric structure well, but that the planetary boundary layer scheme underestimates observed MLH at nighttime. Similarly, this indicates that the mixing might also be underestimated by the boundary layer scheme at nighttime conditions.

Comparing the model results to ceilometer observations from Berlin at the Nansenstraße station (not displayed) also indicates that the diurnal variation is reproduced correctly. The comparison of daily minimum MLH with ceilometer observations also shows an underestimation of MLH-YSU in the same range as at Lindenberg. However, we do not know whether the magnitude of the mixing layer height derived from the ceilometer backscatter profile is directly comparable with the mixing layer height calculated from profiles of temperature, wind speed and humdity, which makes it more difficult to evaluate the modeled mixing layer height quantitatively at the urban site Nansenstraße. For this, further studies assessing the comparability of MLH derived from radiosonde and ceilometer observations would be necessary.

Simulated hourly wind speed correlates with observations with a correlation coefficient between 0.5 and 0.6 (Table S5 in the supplementary material), which is comparable to simulations for the European domain (Mar et al., 2016). Wind speed is overestimated between 0.4 m/s (15%) and 1.4 m/s (50%), depending on the station. The overestimation is especially strong at stations with mean observed wind speeds below 3 m/s, as well as for a period of easterly winds in mid-July (Fig. 7). The most frequently observed wind direction at three stations in Berlin and in Potsdam in June, July and August 2014 is westerly. This is reproduced by the model, with better skill with increasing resolution (Fig. 8). Depending on the modeled wind direction, the bias in wind speed differs: while the bias (averaged over all four stations) is lower than 1m/s for modeled wind from north to south-east, the bias is larger for wind simulated from east and north-east. In addition, the conditional quantile plot of wind speed, split by modeled wind direction, also shows that the model's skill in simulating wind speed from west and south-west is higher (see figure S3 in the supplemenrrary material).

Both the diurnal variability and the magnitude of specific humidity are simulated well by the model, with normalized mean biases between -7% and +7% and correlation coefficients of 3-hourly values of around 0.8 (not shown). Precipitation is simulated well with the 3km and 1km horizontal resolution: both the number of days with precipitation rates larger than 0.01 mm/h and the total amount of precipitation in the simulated period agree well with the observations (Fig. 9). Model results from the 15km resolution overestimate the number of days with precipiation larger than 0.01 mm/h by ca. 30% and the amount by ca. 50%. This shows that the higher resolved domains in the nested setup, using the Grell-Freitas cumulus scheme on all domains, improve the skill in simulating precipitation, which is an important conclusion for future studies with a similar setup aiming at including aqeous phase chemistry and wet scavenging.

## 4.2 Chemistry and aerosols

### 4.2.1 Nitrogen oxides and ozone

The mean bias of modeled $NO_x$ depends on the type of observations that it is compared with (Table 7): for rural sites close to Berlin and Potsdam it is biased positively. Modeled $NO_x$ at urban background sites is mainly biased negatively, while the bias is positive or negative at suburban background sites. The maximum bias of all sites (Table 7) is improved with increasing spatial





resolution from 15km to 3km, from $+11.9\,\mu g\,m^{-3}$ to $+5.3\,\mu g\,m^{-3}$ (rural), $+9.3\,\mu g\,m^{-3}$ to $+6.7\,\mu g\,m^{-3}$ (suburban background), and $-6.7\,\mu g\,m^{-3}$ to $-5.7\,\mu g\,m^{-3}$ (urban background). This indicates that generally a horizontal resolution of 3km is better suited to resolve the spatial $NO_x$ patterns within a city of the size of Berlin even with emission input data coarser than 3km, which is in line with the results of Tie et al. (2010) for Mexico City. A 15 km resolution is not sufficient to resolve the differences between

rural and urban concentrations (Fig. 10). Comparing the mean bias between the 3km and 1km resolutions further shows that, with an emission inventory of 7km horizontal resolution, the 1km resolution does not generally improve the results.

As a first step for model-based assessments of urban $NO_x$ concentrations it is important to be able to simulate daily maximum urban background $NO_x$ concentrations well. In order to assess the model's skill in reproducing these, we compare modeled diurnal cycles of $NO_x$ to observed diurnal cycles (Fig. 11). The comparison shows that the WRF-Chem setup presented here is

not able to simulate the observed diurnal cycle at any of the three resolutions, overestimating $NO_x$ concentrations at nighttime, and underestimating during daytime, not capturing the peak in observed concentrations due to increased traffic densities in the morning and evening hours. The main reason for the nighttime overestimation is likely the model's underestimation of nighttime mixing as discussed above. A contribution to the daytime underestimation might be uncertainties in the emission inventory: while the share of traffic $NO_x$ emissions to total $NO_x$ emissions within Berlin is just above 35% in the TNO-MACC

III inventory, estimates from the Berlin Senate range around 40%-45% for 2008 and 2009 (Berlin Senate Department for Urban Development and the Environment, 2015b). Using an up-to-date bottom up local inventory might contribute to correcting this bias. We can exclude the diurnal cycle applied to the traffic emissions as a reason for the underestimation of the traffic peak in the morning hours - comparing it to diurnal cycles calculated from traffic counts in Berlin it shows a good agreement (Fig. S8 in the supplementary material). An additional source of bias might be the chemical mechanism: in box model studies, Knote

et al. (2015) compared different chemical mechanisms and found a difference in simulated summertime $NO_x$ of up to 25% between the mechanisms. However, the deviation from the multi-mechanism mean was only of the order of a few per cent for summertime conditions simulated with RADM2, which is the mechanism used in this study.

Additionally, we compare the simulated NO and $NO_2$ to observations as described in Sect. 3.1 (Fig. 11, and Fig. S6, S7 and Tables S6, S7 in the supplementary material). As for $NO_x$, the bias of modeled NO depends on the station type. For suburban

and urban background stations NO is on average mainly biased negatively up to $-2.5\,\mu g\,m^{-3}$ (-60%), while it shows a positive bias at some of the rural stations. Part of this negative bias is due to a lower detection limit in the observation data ranging between 0.1 and $2\,\mu g\,m^{-3}$ depending on the station. While this is not the main contribution to the bias in $NO_x$, it does play a larger role when only looking at NO, as for some of the stations a large share of the observed hourly values lies at or below this threshold both in the observed and modeled data (up to 94%). The diurnal cycle of NO is modeled in good agreement with the

observations, but the peak values are underestimated (Fig. 11). Especially for urban sites, the bias is larger when simulated with a 15km resolution than with 3km and 1km resolutions. Modeled $NO_2$ is on average mostly biased high, with up to $11.1\,\mu g\,m^{-3}$, $5.3\,\mu g\,m^{-3}$ and $4.5\,\mu g\,m^{-3}$ for rural sites and up to $10.2\,\mu g\,m^{-3}$, $7.3\,\mu g\,m^{-3}$ and $6.5\,\mu g\,m^{-3}$ for suburban sites (15km, 3km and 1km resolution). Urban background sites are both biased high and low. It is important to note that the posivite bias always results from overestimations during nighttime, while daytime $NO_2$, as total $NO_x$, is always biased low, though with a smaller

daytime bias for suburban and rural site than for the urban background. These results are in line with what has been discussed





for $NO_x$ above and indicate that in addition to the model resolution, the resolution of emissions might play an important role for simulating daytime $NO_x$ concentrations in cities, as more $NO_x$ is emitted near streets than at the edges of the city, which can hardly be captured with emission input data of a horizontal resolution of 7km.

$O_3$ daily means and especially MDA8 ozone are underestimated by the model (Fig. 11 and Table S8), with biases of up to ca. -10 $\mu g\,m^{-3}$ (mean) and -13 $\mu g\,m^{-3}$ (MDA8). This is consistent with what has been reported for a European domain using RADM2 chemistry (Mar et al., 2016) and in line with previous studies showing a deficiency of many online-coupled models, including WRF-Chem with the RADM2 chemical mechanism, in simulating peak ozone concentrations (e.g. Im et al., 2015a). Furthermore, it is in line with studies identifying the choice of chemical mechanism as a reason for differences in simulated ozone concentrations (e.g. Coates and Butler, 2015; Knote et al., 2015). The choice of chemical mechanism, but not so much the modelled meteorology being an important cause of this bias is further supported by the fact that maximum temperatures are generally simulated well by the model, and MDA8 ozone is underestimated even when daily maximum temperatures are simulated correctly. The mean $O_3$ is still simulated reasonably well, though the model underestimates at night and overestimates during the morning hours, which is consistent with a bias in $NO_x$ diurnal cycles discussed above.

### 4.2.2 Particulate matter

The mean bias of the simulated $PM_{10}$ amounts to -50% (Fig. 12 and Table S9 in the supplementary material), which is relatively consistent at all eight stations within and around Berlin as well as at all three model resolutions. Modeled $PM_{2.5}$ concentrations are biased between -20% and -35% (Fig. 12 and Table S10 in the supplementary material). From previous studies with the MADE/SORGAM aerosol scheme it is known that it underestimates the secondary organic aerosol contribution to PM (Ahmadov et al., 2012). Comparing the JJA-averaged model output to components of $PM_{10}$ observed at Nansenstraße during the BAERLIN2014 campaign is in line with these results: while the observations show a mean concentration of organic carbon of 5.6 $\mu g\,m^{-3}$, the modeled particulate organic matter including organic carbon is on average 0.8 $\mu g\,m^{-3}$. In addition, the comparison shows that the contribution of black carbon (BC) to PM might be underestimated, with observed elemental carbon (EC) concentrations of 1.4 $\mu g\,m^{-3}$ on average and mean modeled BC concentrations of 0.2 $\mu g\,m^{-3}$, though the modeled value is still within the range of observed values in individual samples. The underestimation of OC and to a lesser extent BC being a cause of the underestimation of $PM_{10}$ is supported by the fact that, on average, model results compare reasonably well with the observations of other components of $PM_{10}$: modeled sulfate, nitrate and ammonium amounts to 1.8 $\mu g\,m^{-3}$, 0.5 $\mu g\,m^{-3}$ and 0.7 $\mu g\,m^{-3}$, while the mean observed concentrations are 1.9 $\mu g\,m^{-3}$, 0.9 $\mu g\,m^{-3}$ and 0.6 $\mu g\,m^{-3}$. Modeled sea salt amounts to 1.0 $\mu g\,m^{-3}$, and observed sodium and chloride are 0.5 and 0.6 $\mu g\,m^{-3}$, respectively. An additional underestimation of mineral dust or re-suspended road dust emissions, such as brake and tyre wear, primarily contributing to $PM_{10}$, might explain why $PM_{10}$ is underestimated more than $PM_{2.5}$. It should further be noted that the bias of $PM_{2.5}$ daily means varies throughout the simulated period, with the concentrations being biased more negatively in periods where the wind speed is overestimated more strongly. This underlines that the correct simulation of meteorological parameters in the online coupled model WRF-Chem plays an important role in simulating aerosols. The correlation of modeled daily mean $PM_{10}$ concentrations with observations ranges from 0.26 to 0.46 for the 15km resolution, from 0.31 to 0.51 for the 3km resolution and from 0.34 to 0.56 for the



1km resolution. Correlations of simulated $PM_{2.5}$ daily means also fall into this range except at two urban background sites, Brückenstraße and Amrumer Straße, where the correlation coefficient is between 0.17 and 0.26 at all resolutions.

## 5   Sensitivity studies

In this section we address whether the skill in simulating meteorology (T2, WS10, MLH) is improved when updating the urban parameters and specifying land use classes on a sub-grid scale, as well as whether this has an impact on the skill in simulating $NO_x$ concentrations. Furthermore, we analyse whether using a higher resolved emission inventory leads to differences in simulated $NO_x$ concentrations with horizontal model resolutions of 3km and 1km. We focus on $NO_x$, since as mentioned before, the bias found in the base run mean ozone concentrations and maximum daily eight hour ozone are likely not due to the simulated meteorology or resolution of emissions. Similarly, the bias of model results for $PM_{10}$ and $PM_{2.5}$ is mainly due to an underestimation of secondary organic aerosols by the aerosol mechanism as well as missing emissions.

### 5.1   Changes in meteorology in S1_urb and S2_mos

The positive bias in T2 found in the model results at many sites is decreased for urban areas if the input parameters to the urban scheme are specified based on data describing the city of Berlin (simulation S1_urb, Table 4), which is mainly due to the fact that T2 is overall simulated lower for urban areas in this sensitivity simulation. Specifically, there is only one site within the urban area (among all urban built-up and urban green stations) for which the model results with the 1km horizontal resolution (d03) are biased more than $\pm$ 1°C (S1_urb, d02: 3 stations, base run, d03: 3 stations, base run, d02: 6 stations). Likewise, the simulation of daily maximum temperatures is improved. The results from this sensitivity simulation, similarly to the results from the base run, show that the differences between the results of the 3km and 1km resolutions are largest if the urban class of the grid cell changes with changing resolution, though overall the results of the 1km resolution match the observations slightly better than the results obtained with the 3km resolution (Table 4). Even though on average the temperature bias is lower in S1_urb than for the base run, the conditional quantile plots show that the highest observed values are still not captured by the model (Fig. 4).

Using the mosaic option of the land surface scheme and thereby taking into account the subgrid scale variability of the land use classes within one model grid cell (simulation S2_mos) has a similar effect on simulated T2 as in S1_urb: overall, simulated T2 is lower than in the base run, which leads to a decrease in T2 bias compared to observations. Furthermore, it leads to the results from the 1km and 3km resolutions being more similar even at sites with different land use categories, which is refered to as grid convergence by Li et al. (2013) and might indicate that a resolution higher than 3km is not needed in this case. The conditional quantile plots (Fig. 4) underline these results, showing almost identical median values and distributions for the 1km and 3km resolutions, and furthermore reveal that the temperatures simulated with the 15km resolution ressemble the results with 3km and 1km resolutions more than in any of the other simulations. At the 15km model resolution and when applying the mosaic option, gradients at the edges of the city are resolved better than in the other simulations at the 15km resolution, which is expressed through a lower mean bias at sites at the boundaries of Berlin. An important limitation using this option is





the simulated daily maximum T2, which is underestimated at most stations (Table 5). This feature was also found by (Jänicke et al., 2016) for Berlin and surroundings when applying the single-layer urban canopy model in combination with the mosaic approach and indicates that T2 might be decreased too much when using this option.

There is no observational data from radiosondes available within the city, which is why we cannot draw conclusions on the importance of updating the urban parameters or using the mosaic option for urban areas from comparisons with observed profiles of temperatures or MLH. However, knowing that the MLH diagnosed from WRF-Chem (MLH-YSU) is biased low in the base run at nighttime, we compare JJA mean nighttime (20:00-02:00 UTC) MLH from the base run and S1_urb as well as S2_mos (Fig. 13). The results show that the nighttime MLH-YSU is simulated on average up to ca. 30 m lower in S1_urb than in the base run for most grid cells with the land use type low intensity residential. It is simulated higher than in the base run for grid cells with the land use type high intensity residential and commercial, industry, transport. This shows that the urban parameters can strongly influence the meteorology simulated in urban areas and suggests that they might have to be further refined for simulating the urban atmospheric structure correctly.

The nighttime MLH simulated with S2_mos is up to ca. 70m lower than in the base run for urban areas, which is an even larger reduction than in S1_urb. As for S1_urb, grid cells with the dominant urban classes being high intensity residential and commercial/industry/transport have a higher MLH-YSU than other urban grid cells, though this effect is smoothed through the use of the mosaic option.

The bias in wind speed is reduced in S1_urb, ranging from +0.3 m/s (10%) to +1 m/s (34%) depending on the station (Fig. 7, 8 and Table S5). The bias is especially decreased for two periods in mid-June and mid-August, where observed daily mean wind speeds are between 5 and 6 m/s, which is relatively high compared to the rest of the simulated period. In the base run, the model overestimates the observations during these periods, which is not the case in S1_urb. Similarly, the wind speeds during the periods in mid-July with easterly wind, where the base run strongly overestimates wind speeds, are biased by ca. 1-2 m/s less (Fig. 7). The histograms in the conditional quantile plots further shows that the range of modeled wind speeds from S1_urb matches the range of observed wind speed better than in the base run (Fig. S4 in the supplementary material).

Similar to S1_urb, the bias in wind speed is decreased in S2_mos, ranging from below +0.1 m/s (2%) to +1.2 m/s (40%) (Fig. 7, 8, and Table S5, Fig. S4 in the supplementary material). However, it should be noted that unlike for S1_urb, where the decrease in wind speed is distributed evenly throughout the day, wind speed in S2_mos is especially lower at nighttime, while maximum diurnal wind speeds are similar to those simulated in the base run (not shown).

Overall, the results show that when using a model setup with highly resolved nests the simulated meterology seems to be improved both by specifying land use input data and urban parameters for the simulated region and when using the mosaic option, though diurnal cycles of T2 and wind speed are improved more in S1_urb. Particularly the differences between S1_urb and the base run for grid cells with land use types high intensity residential and industry/commercial/transport reveal that the specification of urban parameters can contribute to improving the model bias also in MLH. The results from S2_mos show that the mosaic option might be a useful alternative if computational resources are too limited to include higher resolved nested domains.





## 5.2 Impact of meteorology changes on simulated $NO_x$ concentrations

Mean $NO_x$ concentrations simulated with S1_urb are generally higher than those simulated with the base run, with the difference between S1_urb and the base run for grid cells of the measurement stations of up to 9% (15km resolution), up to 13% (3km resolution), and up to 18% (1km resolution). Thus, the positive bias which has been found in the base run is increased in S1_urb. For all three domains, the differences are larger for urban grid cells. An analysis of the diurnal cycles reveals that these differences are mainly due to higher nighttime $NO_x$ concentrations in S1_urb (Fig. 11). This is consistent with previous results: an underestimation of MLH by the model (MLH-YSU) at nighttime leads to an overestimation of $NO_x$. An even lower MLH in this sensitivity simulation (section 5.1) explains nighttime $NO_x$ concentrations being higher than in the base run. The overestimation of nighttime $NO_x$ might be further reinforced by lower simulated wind speeds in S1_urb. Daytime $NO_x$, which we define as $NO_x$ concentrations between 7:00 and 17:00 UTC, changes only little in S1_urb compared with the base run at urban background stations in Berlin: results with a 3km horizontal resolution show an increase in daytime $NO_x$ in S1_urb between 2% and and 5%, and an increase between 5% and 7% with a 1km resolution compared to the base run.

Results for simulated $NO_x$ from S2_mos are consistent with the results from S1_urb: simulated nighttime $NO_x$ is even higher that that simulated in the base run and in S1_urb, which is consistent with the larger difference between MLH-YSU simulated with the base run settings and within S2_mos. Daytime $NO_x$ changes even less in S2_mos compared to the base run, with changes between -1% and +2% (3km resolution) or +3% to +5% (1km resolution).

Overall, the results underline that the underestimation of mixing in the boundary layer is likely to have a strong influence on simulated nighttime $NO_x$ concentrations in urban areas, which is not corrected using the mosaic option or specifying the input parameters to the urban scheme. However, since the simulated MLH is sensitive to the change in urban parameters for high intensity residential and commercial/industry/transport urban areas, it shows that this could potentially have an impact on simulated $NO_x$ concentrations. The results from both S1_urb and S2_mos show that daytime $NO_x$ is influenced little by changes in the modeled meteorology, suggesting that the bias in daytime $NO_x$ is due to too low emissions or an incorrect distribution of emissions resulting from a too coarse resolution of the emission inventory as mentioned in section 4.2.

## 5.3 Resolution of the emission inventory

Evaluating the base run (Sect. 4) we found that the improvement in simulating $NO_x$ concentrations with a 1km horizontal resolution, as compared to a horizontal resolution of 3km, is negligible when using emission input data at 7km horizontal resolution. This result changes when providing emission input data with a horizontal resolution of ca. 1km as described in section 2.4 (Fig. 10): the model is then able to resolve small scale air pollution patterns and hotspots, which cannot be resolved at a horizontal resolution of 3km. A comparison of the results for the urban background stations within Berlin (Amrumer Straße, Belziger Straße, Nansenstraße, Johanna und Willi Brauer Platz, Brückenstraße) helps to illustrate this: in order to minimize the bias by too little nighttime mixing, we only compare daytime (7:00 - 17:00 UTC) $NO_x$ simulated with 3km and 1km horizontal resolution and downscaled emissions. Going from a 3km to a 1km resolution, daytime $NO_x$ changes by +40%, +12%, -25%, +16% and +161% in S3_emi for the above-mentioned urban background sites, respectively (Fig. 11). As a comparison, the



respective changes from the base run are +3%, +1%, -8%, -3% and -3%. This shows that a 1km horizontal model resolution only leads to different results from a 3km horizontal resolution when also using highly resolved emission input data.

Furthermore, the results from the above-mentioned urban background stations show that too low emissions within the city (either due to too low emissions overall or locally too low emissions because of a coarse resolution of the emission inventory)
can be a cause of the bias in daytime $NO_x$ concentrations. To illustrate that, we compare the daytime $NO_x$ concentrations from the base run and S3_emi. Using the original emissions, the emissions summed up over JJA in the the grid cell where the respective station is located are $7.0\,t\,km^{-2}$, $5.4\,t\,km^{-2}$, $6.9\,t\,km^{-2}$, $3.1\,t\,km^{-2}$ and $7.0\,t\,km^{-2}$ for the above-mentioned urban background stations, respectively, and $22.4\,t\,km^{-2}$, $8.4\,t\,km^{-2}$, $6.2\,t\,km^{-2}$, $2.5\,t\,km^{-2}$ and $79.9\,t\,km^{-2}$ in the downscaled emission data. It should, however, be noted that, though downscaling of the original emissions can lead to a decrease in emission
strength in some of the urban grid cells, it generally results in an increase in the city center and a decrease in the suburban areas. This is due to the population density and the traffic density, which are used as proxies for the emission downscaling, being higher in the city center. Using the downscaled emission data leads to an increase in simulated daytime $NO_x$ of 23%, 22%, 52%, 20% and 51% (3km resolution) or 68%, 36%, 24%, 44% and 308% (1km resolution) at the above-mentioned urban background stations, as compared to the base run. This shows that, despite small decreases in emissions in some of the grid
cells, the generally increased $NO_x$ emissions in the city center lead to increases in simulated $NO_x$ concentration at all five sites. This result indicates that the downscaled emissions might be more suitable to represent gradients in emissions in the urban area, contributing to correcting the bias in simulated daytime urban $NO_x$ in the base run.

A comparison of results from S3_emi with observations at Brückenstraße (Table 7) shows that locally the bias in simulated $NO_x$ concentrations can increase strongly. While for most urban background stations in Berlin using the downscaled emissions
improves both the bias of mean $NO_x$ and the bias of daytime $NO_x$, the example of Brückenstraße shows that further modifications to the emission downscaling and processing might be necessary when simulating local $NO_x$ patterns: at the Brückenstraße site, the mean bias increases from -4 $\mu g\,m^{-3}$ (1km resolution, base run) to +26 $\mu g\,m^{-3}$ (1km resolution, S3_emi). The large overestimation is due to a point source being close to the site and the way point sources have been treated: as mentioned in section 2.4, point source emissions are all released into the first model layer. Furthermore, the point source emissions are dis-
tributed as area sources at the resolution of the emission inventory. This results in much higher emissions over a much smaller area in the downscaled emission inventory, locally increasing the concentrations in the vicinity of point sources. Likewise, the comparison of simulated and observed concentrations at rural and suburban sites just outside of Berlin shows that the model skill suffers from the lack of proxy data specifying the spatial distribution of emissions directly outside of Berlin.

Generally, comparing the results from the base run with the results from S3_emi leads to several conclusions: when simu-
30 lating $NO_x$ concentrations in urban areas, a horizontal model resolution can be beneficial if an emission inventory of similarly high resolution is available. However, using a highly resolved emission inventory for a model domain with a similarly high resolution is only beneficial for improving the comparability with observations and the application to local studies if the emission inventory is of sufficient spatial precision. The downscaling approach presented here shows how locally highly resolved emissions can be calculated effectively and consistently by combining a readily available emission inventory with data available
for many urban areas, such as population and traffic densities. Our results suggest that a further refinement of the proxy data



could be useful, e.g. using proxy datasets covering more than the urban area itself. Further refinements could consist in using the housing type (or high population density as an indication for high-rise buildings) for better distributing residential heating emissions. As for the vertical distribution of emissions, Mar et al. (2016) state it has little impact on the model results. While this might hold for simulations of rural background air quality with domain resolutions of the order of 45km, the present results suggest that it is of higher relevance to distribute point source emissions into several vertical model levels when decreasing the model resolution and the resolution of the emission input data.

## 6 Summary and conclusions

In this study, we evaluate a WRF-Chem setup for the Berlin-Brandenburg area with three nested model domains of 15km, 3km and 1km horizontal resolutions for three months in summer 2014. The results show that the model generally simulates meteorology well, though urban 2m temperature and urban wind speeds are biased high and nighttime mixing layer height is biased low in the base run. On average, ozone is simulated reasonably well, but maximum daily eight hour mean concentrations are underestimated, which is consistent with the results from previous modelling studies using the RADM2 chemical mechanism. Particulate matter is underestimated, which is at least partly explained by an underestimation of secondary organic aerosols and consistent with previous studies. $NO_x$ concentrations are simulated reasonably well on average, but overestimated at nighttime and underestimated at daytime especially in the urban areas.

We specifically assess how the skill in simulated $NO_x$ is influenced by the model resolution, the prescribed emissions and the simulated meteorology, in turn depending on the model resolution, land use input data to the model, and on the parametrization of the urban structure. This is done with three sensitivity simulations, including updating the representations of the urban structure within the urban canopy model (S1_urb), taking into account a sub-grid scale parametrization of the land use classes (S2_mos), and downscaling the original emission input data from a horizontal resolution of ca. 7km to ca. 1km (S3_emi).

For the base model run, a horizontal resolution of 1km did not generally improve the results compared to a model resolution of 3km. Furthermore, the mosaic option of the Noah land use model, enabling a sub-grid-scale parametrization of the land use classes, lead to a convergence of the results at the different model resolutions rather than an improvement of the results at the 1km model resolution. However, this study has shown that a 1km horizontal model resolution can be very valuable for simulating urban background air quality in the Berlin-Brandenburg region with small modifications, including a better representation of the nighttime mixing layer height in the model, a more detailed specification of urban land use together with the respective input parameters to the urban canopy model and a better spatial representation of urban emissions.

The simulation of the urban boundary layer height is crucial for correctly simulating diurnal cycles of $NO_x$. In the base run, daily minimum (nighttime) mixing layer height simulated by the model is lower than observations outside of the urban area by more than 50% on all domains. This is consistent with a strong modeled overestimation of $NO_x$ at nighttime. However, when calculating the mixing layer height from modeled profiles of temperature, wind speed and humidity the nighttime bias decreases to ca. +8% to ca. 26%. Daily maximum mixing layer height is biased less, and the difference is smaller between the



two different approaches of calculating the mixing layer height. This indicates that the calculation of the urban boundary layer height and nighttime mixing in the model might need to be adapted to better represent observed conditions at nighttime.

A more detailed specification of urban land use classes together with the respective input parameters can help better represent the heterogeneity of urban area in a model domain with 1km horizontal resolution. This is shown by the modeled 2m temperature only differing by more than 0.1°C between the model resolutions of 3km and 1km if the land use class of the respective grid cell changes. It is further shown by the simulation with updated urban parameters decreasing the positive bias in simulated wind speed in the base run by up to 0.5 m/s, from a mean bias in wind speed up to 1.5 m/s in the base run to a mean bias in wind speed of maximally 1m/s in the sensitivity simulation where urban parameters have been updated. In addition, the nighttime mixing layer height is simulated higher in this sensitivity simulation for grid cells of the urban types high intensity residential and commercial/industry/transport, suggesting that the negative bias in mixing layer height at nighttime can also be corrected by better specifying the input parameters to the urban scheme and the urban land use classes.

When downscaling the emissions from a horizontal resolution of 7km to 1km based on proxy data for Berlin including population density and traffic densities, local pollution patterns can be resolved better with a model domain with a horizontal resolution of 1km, compared to 3km. A particular strength of this approach is its effective and consistent combination of a readily available emission inventory and locally available data, which can be applied generically to urban areas. In order to further refine this approach, the downscaling of the coarse emission inventory could be extended especially at and beyond the boundaries of the urban area, or the proxy data for industrial and residential heating emissions could be further refined. Alternatively, a highly resolved local bottom-up emission inventory can help increase the model's skill when simulating with a horizontal resolution of 1km. In addition, the results have shown that a more detailed treatment of point source emissions including their vertical distribution becomes important when going to a horizontal model resolution of 1km.

Overall, these results can build a basis for the design of future air quality modelling studies over the Berlin-Brandenburg region and other European urban agglomerations of similar extent. The above-mentioned suggested modifications to the setup are based on data which, to a large extent, is available or easily producable for the Berlin-Brandenburg region and other European urban areas. Considering these modifications we find the presented WRF-Chem configuration at a 1km horizontal resolution a suitable setup for simulating urban background $NO_x$ concentrations, when used together with the single layer urban canopy model with input parameters specified for the city of interest and combined with emission input data of a similar resolution as the model domain.

## 7   Code availability

WRF-Chem is an open-source, publicly available community model. A new, improved version is released approximately twice a year. The WRF-Chem code is available at http://www2.mmm.ucar.edu/wrf/users/download/get_source.html. The corresponding author will provide the modifications introduced and described in Sect. 2 upon request.



*Acknowledgements.* The authors would like to thank Andreas Kerschbaumer (Berlin Senate Department for Urban Development and the Environment) for providing data on the building structure and land use of Berlin, as well as for valuable discussions and input on using the data for this study. In addition, further data on the popultation density, traffic density and road network of Berlin has been obtained through the Environment Data Base of the Berlin Senate Department for Urban Development and the Environment. We thank Erika von Schneidemesser

and Boris Bonn for providing measurement data from the BAERLIN2014 campaign as well as for valuable discussions of the data and results. We thank Noelia Otero for providing the algorithm on the calculation of weather types. We thank Georg Grell and his colleagues for discussions of the WRF-Chem setup. We would further like to thank Renate Forkel and Joachim Fallmann for valuable discussions regarding the setup and results of our WRF-Chem simulation. We thank TNO for access to the TNO-MACC III emissions inventory. We acknowledge the UK Met Office for providing the Global Weather Observation dataset via the British Atmospheric Data Centre. We acknowledge the Ger-

man Federal Environment Agency and the Berlin Senate Department for Urban Developement for providing air quality observations from the Federal States' networks, including the BLUME network in Berlin. Mixing layer heights calculated from radiosonde observations in Lindenberg have kindly been provided by F. Beyrich (DWD). WRF-Chem tools for preprocessing boundary conditions as well as biogenic and anthropogenic emissions were provided by NCAR (http://www.acom.ucar.edu/wrf-chem/download.shtml). Initial and boundary conditions for meteorological fields were obtained from ECMWF, http://www.ecmwf.int/en/research/climate-reanalysis/era-interim. Initial and bound-

ary conditions for chemical fields were from MOZART-4/GEOS5, provided by NCAR at http://www.acd.ucar.edu/wrf-chem/mozart.shtml. Corine land cover data was obtained from http://www.eea.europa.eu/data-and-maps/data/corine-land-cover-2006-raster-2. The data analysis has been done with the open source software R, including its library openair (R Core Team, 2013; Carslaw and Ropkins, 2012) as well as with the NCAR command language (UCAR/NCAR/CISL/TDD, 2016). The WRF-Chem simulations were done on the high performance cluster of the Potsdam Institute for Climate Impact Research.





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



**Table 1.** Physics and chemistry parametrization.

| Process | Scheme | Remarks |
|---|:---:|---:|
| Cloud micropysics | Morrison double-moment | |
| Radiation (short wave) | RRTMG | called every 15 minutes |
| Radiation (long wave) | RRTMG | called every 15 minutes |
| Boundary layer physics | YSU | |
| Urban scheme | single-layer urban canopy model | 3 categories: roofs, walls, streets |
| Land surface processes | Noah LSM | CORINE land use input data |
| Cumulus convection | Grell-Freitas | switched on for all domains |
| Chemistry | RADM2 | KPP version (chem_opt=106) |
| Aerosol particles | MADE/SORGAM | |
| Photolysis | Madronich F-TUV | |



**Table 2.** Urban parameters for Berlin for the three urban classes low intensity residential (31), high intensity residential (32) and commercial, industry, transport (33)

| Parameter | Default (class 31 / 32 / 33) | Updated (class 31 / 32 / 33) |
|---|---|---|
| Roof level (m) | 5 / 7.5 / 10 | 3 / 15 / 3 |
| Standard deviation of roof height (m) | 1 / 3 / 4 | 4.4 / 6.3 / 5.2 |
| Roof width (m) | 8.3 / 9.4 / 10 | 8.3 / 16.0 / 11.8 |
| Road width (m) | 8.3 / 9.4 / 10 | 17.5 / 17.5 / 17.5 |
| Fraction of urban landscape without natural vegetation | 0.5 / 0.9 / 0.95 | 0.4 / 0.7 / 0.48 |





**Table 3.** Observational data in Berlin and Potsdam. If one class is given, it refers to the meteorology class if the network is DWD, GUAN or TU, and to the chemisty class otherwise. The abbreviated name (Abbr.) is referred to in tables sumarizing statistics for the different stations.

| Station | Abbr. | Network | Class (meteorology/chemistry) | Species used |
|---|---|---|---|---|
| Nansenstraße | nans | BÄRLIN | urban built-up / urban background | $PM_{2.5}$, PM comp., MLH |
| Nansenstraße | nans | BLUME | urban built-up / urban background | NO, $NO_2$, $O_3$, $PM_{10}$, T2 |
| Amrumer Straße | amst | BLUME | urban background | $PM_{10}$, $PM_{2.5}$, NO, $NO_2$, $O_3$ |
| Belziger Strße | belz | BLUME | urban background | $PM_{10}$, NO, $NO_2$ |
| Brueckenstraße | brue | BLUME | urban background | $PM_{10}$, $PM_{2.5}$, NO, $NO_2$ |
| J. u. W. Brauer Platz | jwbp | BLUME | urban background | NO, $NO_2$ |
| Potsdam-Zentrum | pots | UBA | urban background | $PM_{10}$, $PM_{2.5}$, NO, $NO_2$, $O_3$ |
| Blankenfelde-Mahlow | blan | UBA | suburban background | $PM_{10}$, $PM_{2.5}$, NO, $NO_2$, $O_3$ |
| Buch | buch | BLUME | suburban background | $PM_{10}$, NO, $NO_2$, $O_3$ |
| Grunewald | grun | BLUME | suburban background | $PM_{10}$, NO, $NO_2$, $O_3$ |
| Potsdam, Groß Glienicke | glie | UBA | suburban background | $PM_{10}$, NO, $NO_2$, $O_3$ |
| Schichauweg | schw | BLUME | rural industrial | NO, $NO_2$, $O_3$ |
| Mueggelseedamm | mueg | BLUME | rural background | $PM_{10}$, NO, $NO_2$, $O_3$ |
| Frohnau | froh | BLUME | rural background | NO, $NO_2$, $O_3$ |
| Marzahn | marz | DWD | urban built-up | T2, prec., Q2 |
| Botanischer Garten | botg | DWD/FU | urban green | T2, prec., Q2 |
| Tegel | tege | DWD | urban green | T2, WS10, WD10, prec., Q2 |
| Tempelhof | temp | DWD | urban green | T2, WS10, WD10, prec., Q2 |
| Buch | buch | DWD | urban green | T2, prec., Q2 |
| Kaniswall | kani | DWD | rural | T2, prec., Q2 |
| Potsdam | pots | DWD | rural | T2, WS10, WD10, prec., Q2 |
| Schoenefeld | scho | DWD | rural | T2, WS10, WD10, prec., Q2 |
| Lindenberg | lind | DWD/GRUAN | rural | T, ws, q profiles |
| Bamberger Straße | bamb | TU | urban built-up | T2 |
| Dessauer Straße | dest | TU | urban built-up | T2 |
| Rothenburgstraße | roth | TU | urban built-up | T2 |
| Albrechtstrasse | albr | TU | urban green | T2 |
| Tiergarten | tier | TU | urban green | T2 |
| Dahlemer Feld | dahf | TU | rural | T2 |





**Table 4.** Statistics of hourly 2m temperature for JJA for stations, where the land use class of the respective grid cell changes with resolution. „LU" refers the WRF land use class of the grid cell in the respective domain, „Obs" refers to the JJA observed mean, „Mod" refers to the JJA modeled mean for the respective grid cell. MB is the mean bias for JJA and r is the correlation of hourly values. Obs, Mod and MB are in °C. The statistics are shown for the results from the model domains of 15km (d01), 3km (d02) and 1km (d03) horizontal resolution.

| Station | | LU | Obs | base Mod | MB | r | S1_urb Mod | MB | r | S2_mos Mod | MB | r |
|---|---|---|---|---|---|---|---|---|---|---|---|---|
| kani | d01 | 31 | 18.1 | 19.6 | 1.5 | 0.88 | 19.3 | 1.2 | 0.88 | 19.2 | 1 | 0.89 |
| | d02 | 2 | 18.1 | 19.4 | 1.3 | 0.9 | 19.3 | 1.2 | 0.9 | 19.3 | 1.1 | 0.89 |
| | d03 | 2 | 18.1 | 19.4 | 1.2 | 0.9 | 19.2 | 1.1 | 0.9 | 19.2 | 1.1 | 0.89 |
| marz | d01 | 2 | 19.2 | 18.8 | -0.4 | 0.91 | 18.7 | -0.6 | 0.9 | 18.9 | -0.4 | 0.92 |
| | d02 | 31 | 19.2 | 19.6 | 0.4 | 0.91 | 19.4 | 0.2 | 0.9 | 19.2 | 0 | 0.9 |
| | d03 | 31 | 19.2 | 19.7 | 0.4 | 0.91 | 19.3 | 0.1 | 0.9 | 19.2 | 0 | 0.9 |
| scho | d01 | 31 | 18.8 | 19.6 | 0.8 | 0.92 | 19.3 | 0.6 | 0.91 | 19.2 | 0.4 | 0.92 |
| | d02 | 31 | 18.8 | 19.9 | 1.1 | 0.91 | 19.7 | 0.9 | 0.91 | 19.4 | 0.6 | 0.91 |
| | d03 | 2 | 18.8 | 19.3 | 0.6 | 0.92 | 19.2 | 0.4 | 0.91 | 19.3 | 0.6 | 0.91 |
| temp | d01 | 31 | 19.3 | 19.6 | 0.3 | 0.92 | 19.3 | 0 | 0.91 | 19.3 | -0.1 | 0.92 |
| | d02 | 33 | 19.3 | 20.3 | 0.9 | 0.9 | 19.7 | 0.4 | 0.9 | 19.6 | 0.3 | 0.9 |
| | d03 | 33 | 19.3 | 20.2 | 0.8 | 0.9 | 19.6 | 0.3 | 0.9 | 19.5 | 0.2 | 0.9 |
| nans | d01 | 31 | 20.8 | 19.6 | -1.1 | 0.91 | 19.3 | -1.4 | 0.9 | 19.3 | -1.5 | 0.91 |
| | d02 | 31 | 20.8 | 19.9 | -0.9 | 0.9 | 19.6 | -1.1 | 0.89 | 19.6 | -1.2 | 0.9 |
| | d03 | 32 | 20.8 | 20.2 | -0.5 | 0.9 | 20 | -0.8 | 0.89 | 19.6 | -1.2 | 0.9 |
| dahf | d01 | 31 | 17.9 | 19.6 | 1.6 | 0.88 | 19.3 | 1.4 | 0.89 | 19.1 | 1.1 | 0.9 |
| | d02 | 14 | 17.9 | 19.3 | 1.4 | 0.9 | 19.1 | 1.2 | 0.9 | 19.3 | 1.4 | 0.88 |
| | d03 | 14 | 17.9 | 19.2 | 1.3 | 0.9 | 19 | 1.1 | 0.9 | 19.2 | 1.3 | 0.88 |
| bamb | d01 | 31 | 19.3 | 19.6 | 0.4 | 0.9 | 19.3 | 0.1 | 0.89 | 19.3 | 0 | 0.91 |
| | d02 | 31 | 19.3 | 19.9 | 0.6 | 0.89 | 19.6 | 0.4 | 0.88 | 19.6 | 0.3 | 0.9 |
| | d03 | 32 | 19.3 | 20.2 | 0.9 | 0.9 | 19.9 | 0.7 | 0.89 | 19.5 | 0.2 | 0.9 |





**Table 5.** Statistics of daily maximum 2m temperature for JJA for stations, where the land use class of the respective grid cell changes with resolution. „LU" refers the WRF land use class of the grid cell in the respective domain, „Obs" refers to the JJA observed mean, „Mod" refers to the JJA modeled mean for the respective grid cell. MB is the mean bias for JJA and r is the correlation of hourly values. Obs, Mod and MB are in °C. The statistics are shown for the results from the model domains of 15km (d01), 3km (d02) and 1km (d03) horizontal resolution.

| Station | | | | base | | | S1_urb | | | S2_mos | | |
|---|---|---|---|---|---|---|---|---|---|---|---|---|
| | | LU | Obs | Mod | MB | r | Mod | MB | r | Mod | MB | r |
| kani | d01 | 31 | 24.2 | 23.8 | -0.4 | 0.88 | 23.6 | -0.6 | 0.87 | 23.3 | -0.9 | 0.89 |
| | d02 | 2 | 24.2 | 24.4 | 0.2 | 0.9 | 24.3 | 0.1 | 0.87 | 23.9 | -0.3 | 0.9 |
| | d03 | 2 | 24.2 | 24.3 | 0.1 | 0.9 | 24.2 | 0 | 0.87 | 23.8 | -0.4 | 0.89 |
| marz | d01 | 2 | 23.9 | 23.4 | -0.5 | 0.88 | 23.2 | -0.8 | 0.86 | 23 | -1 | 0.9 |
| | d02 | 31 | 23.9 | 24.2 | 0.2 | 0.89 | 24 | 0 | 0.87 | 23.5 | -0.4 | 0.9 |
| | d03 | 31 | 23.9 | 24.1 | 0.2 | 0.89 | 23.9 | 0 | 0.87 | 23.5 | -0.5 | 0.9 |
| scho | d01 | 31 | 23.8 | 23.8 | 0 | 0.88 | 23.6 | -0.3 | 0.87 | 23.3 | -0.5 | 0.9 |
| | d02 | 31 | 23.8 | 24.4 | 0.6 | 0.9 | 24.3 | 0.5 | 0.88 | 23.8 | 0 | 0.91 |
| | d03 | 2 | 23.8 | 24.3 | 0.5 | 0.9 | 24.1 | 0.3 | 0.88 | 23.7 | -0.1 | 0.9 |
| temp | d01 | 31 | 24.1 | 23.8 | -0.3 | 0.88 | 23.5 | -0.6 | 0.87 | 23.3 | -0.8 | 0.89 |
| | d02 | 33 | 24.1 | 24.5 | 0.3 | 0.9 | 24.3 | 0.2 | 0.87 | 23.8 | -0.3 | 0.9 |
| | d03 | 33 | 24.1 | 24.4 | 0.2 | 0.9 | 24.2 | 0 | 0.87 | 23.6 | -0.5 | 0.91 |
| nans | d01 | 31 | 25.5 | 23.8 | -1.7 | 0.86 | 23.5 | -1.9 | 0.85 | 23.3 | -2.2 | 0.88 |
| | d02 | 31 | 25.5 | 24.4 | -1.1 | 0.87 | 24.2 | -1.3 | 0.85 | 23.8 | -1.7 | 0.88 |
| | d03 | 32 | 25.5 | 24.5 | -1 | 0.87 | 24.2 | -1.3 | 0.85 | 23.6 | -1.8 | 0.88 |
| dahf | d01 | 31 | 23.8 | 23.7 | -0.1 | 0.89 | 23.5 | -0.3 | 0.88 | 23.3 | -0.5 | 0.9 |
| | d02 | 14 | 23.8 | 24.1 | 0.3 | 0.9 | 24 | 0.2 | 0.88 | 23.7 | -0.1 | 0.9 |
| | d03 | 14 | 23.8 | 24 | 0.2 | 0.9 | 23.8 | 0 | 0.88 | 23.5 | -0.3 | 0.9 |
| bamb | d01 | 31 | 22.9 | 23.8 | 0.9 | 0.88 | 23.5 | 0.7 | 0.87 | 23.3 | 0.4 | 0.9 |
| | d02 | 31 | 22.9 | 24.4 | 1.5 | 0.89 | 24.2 | 1.3 | 0.87 | 23.8 | 0.9 | 0.9 |
| | d03 | 32 | 22.9 | 24.4 | 1.5 | 0.9 | 24.1 | 1.2 | 0.87 | 23.6 | 0.7 | 0.9 |





**Table 6.** Statistics of daily minimum, mean and maximum mixing layer height for JJA. „Obs" refers to the JJA observed mean, „Mod" refers to the JJA modeled mean for the respective grid cell. MB is the mean bias for JJA, NMB refers to the normalized mean bias and r is the correlation of hourly values. The values given in the column „YSU"refer to the MLH diagnosed directly by WRF-Chem, while „calc"refers to the MLH calculated from modeled profiles of temperature, wind speed and humidity. Obs, Mod and MB are given in meters and NMB is given in %. The statistics are shown for the results from the model domains of 15km (d01), 3km (d02) and 1km (d03) horizontal resolution.

| Station | | | | YSU | | | | calc | | | |
|---|---|---|---|---|---|---|---|---|---|---|---|
| | | | Obs | Mod | MB | NMB | r | Mod | MB | NMB | r |
| **Lindenberg** | max | d01 | 1414.1 | 1657.9 | 243.8 | 17.2 | 0.29 | 1681.8 | 267.7 | 18.9 | 0.28 |
| | | d02 | 1414.1 | 1701.5 | 287.3 | 20.3 | 0.22 | 1761.1 | 347 | 24.5 | 0.2 |
| | | d03 | 1414.1 | 1635.4 | 221.3 | 15.6 | 0.21 | 1708.8 | 294.7 | 20.8 | 0.19 |
| | mean | d01 | 689.8 | 736.3 | 46.6 | 6.8 | 0.33 | 777.2 | 87.4 | 12.7 | 0.27 |
| | | d02 | 689.8 | 718.7 | 28.9 | 4.2 | 0.28 | 802.8 | 113 | 16.4 | 0.22 |
| | | d03 | 689.8 | 685.7 | -4 | -0.6 | 0.27 | 783.3 | 93.5 | 13.6 | 0.22 |
| | min | d01 | 187.5 | 88.8 | -98.7 | -52.6 | 0.09 | 202.1 | 14.6 | 7.8 | 0.26 |
| | | d02 | 187.5 | 74.4 | -113.1 | -60.3 | 0.07 | 228.8 | 41.4 | 22.1 | 0.27 |
| | | d03 | 187.5 | 75 | -112.4 | -60 | 0.17 | 235.8 | 48.4 | 25.8 | 0.31 |
| **Nansenstraße** | max | d01 | 2312.8 | 1672.2 | | | | 1701.4 | | | |
| | | d02 | 2312.8 | 1792.7 | | | | 1825.8 | | | |
| | | d03 | 2312.8 | 1760.6 | | | | 1787.2 | | | |
| | mean | d01 | 906.7 | 774 | | | | 825.6 | | | |
| | | d02 | 906.7 | 785.2 | | | | 843.9 | | | |
| | | d03 | 906.7 | 741.4 | | | | 843.7 | | | |
| | min | d01 | 175.4 | 93.7 | | | | 210.1 | | | |
| | | d02 | 175.4 | 76.9 | | | | 197.2 | | | |
| | | d03 | 175.4 | 53.1 | | | | 212.3 | | | |





**Table 7.** Statistics of daily $NO_x$ for JJA. „Obs" refers to the JJA observed mean, „mod" refers to the JJA modeled mean for the respective grid cell. MB is the mean bias for JJA, NMB refers to the normalized mean bias and r is the correlation of hourly values. Obs, Mod and MB are given in $\mu g\,m^{-3}$ and NMB is given in %. The statistics are shown for the results from the model domains of 15km (d01), 3km (d02) and 1km (d03) horizontal resolution.

| St. | | | base | | | | S1_urb | | | | S2_mos | | | | S3_mos | | | |
|---|---|---|---|---|---|---|---|---|---|---|---|---|---|---|---|---|---|---|
| | | Obs | Mod | MB | NMB | r | Mod | MB | NMB | r | Mod | MB | NMB | r | Mod | MB | NMB | r |
| froh | d01 | 8.3 | 20.2 | 11.9 | 143.7 | 0.56 | 22 | 13.7 | 164.6 | 0.43 | 26 | 17.7 | 213.2 | 0.55 | 18.4 | 10.1 | 121.2 | 0.45 |
| | d02 | 8.3 | 10.3 | 2 | 24.6 | 0.55 | 10.6 | 2.3 | 28.1 | 0.48 | 11.4 | 3.1 | 37.1 | 0.55 | 8.4 | 0.1 | 1.6 | 0.5 |
| | d03 | 8.3 | 10.1 | 1.8 | 21.4 | 0.56 | 10.7 | 2.4 | 28.5 | 0.49 | 10.7 | 2.4 | 29.3 | 0.56 | 8.2 | -0.1 | -0.8 | 0.49 |
| grun | d01 | 9.1 | 12.4 | 3.3 | 36.2 | 0.46 | 13.1 | 4 | 43.7 | 0.46 | 16.4 | 7.3 | 80 | 0.49 | 9.3 | 0.2 | 1.7 | 0.42 |
| | d02 | 9.1 | 16.1 | 7 | 76.6 | 0.3 | 16.7 | 7.6 | 83.2 | 0.38 | 18.4 | 9.3 | 101.7 | 0.39 | 12.8 | 3.7 | 40.2 | 0.42 |
| | d03 | 9.1 | 15.8 | 6.7 | 73.8 | 0.27 | 16.5 | 7.3 | 80.5 | 0.37 | 18.7 | 9.6 | 104.9 | 0.33 | 11.6 | 2.5 | 27.3 | 0.31 |
| mueg | d01 | 9.1 | 14 | 4.9 | 53.7 | 0.42 | 15.4 | 6.2 | 68.1 | 0.36 | 17.7 | 8.6 | 94.2 | 0.49 | 12.1 | 3 | 32.9 | 0.37 |
| | d02 | 9.1 | 14.4 | 5.3 | 58 | 0.4 | 16.2 | 7.1 | 77.5 | 0.36 | 16.7 | 7.5 | 82.6 | 0.5 | 13.2 | 4.1 | 44.8 | 0.33 |
| | d03 | 9.1 | 13.5 | 4.3 | 47.6 | 0.45 | 14.7 | 5.5 | 60.4 | 0.38 | 15.3 | 6.2 | 67.6 | 0.52 | 12.1 | 3 | 32.7 | 0.37 |
| schw | d01 | 11.7 | 21.8 | 10.1 | 86 | 0.41 | 23.3 | 11.6 | 98.7 | 0.34 | 27.4 | 15.7 | 133.6 | 0.49 | 20.5 | 8.8 | 74.8 | 0.31 |
| | d02 | 11.7 | 14.2 | 2.5 | 20.9 | 0.36 | 15.2 | 3.5 | 29.7 | 0.36 | 16.3 | 4.6 | 38.9 | 0.48 | 10.5 | -1.3 | -10.9 | 0.2 |
| | d03 | 11.7 | 14 | 2.3 | 19.3 | 0.39 | 15.4 | 3.6 | 31.1 | 0.38 | 16 | 4.3 | 36.8 | 0.47 | 11.3 | -0.4 | -3.2 | 0.18 |
| blan | d01 | 11.9 | 10.8 | -1.1 | -9.2 | 0.26 | 10.7 | -1.2 | -10 | 0.2 | 10.8 | -1 | -8.6 | 0.24 | 9.6 | -2.2 | -18.8 | 0.17 |
| | d02 | 11.9 | 12.8 | 0.9 | 7.6 | 0.22 | 13.7 | 1.8 | 15.5 | 0.21 | 14.8 | 2.9 | 24.6 | 0.31 | 10.4 | -1.5 | -12.4 | 0.17 |
| | d03 | 11.9 | 11.2 | -0.6 | -5.4 | 0.26 | 11.7 | -0.2 | -1.8 | 0.18 | 12.5 | 0.7 | 5.6 | 0.29 | 9.1 | -2.7 | -23.1 | 0.16 |
| buch | d01 | 11 | 20.2 | 9.3 | 84.2 | 0.62 | 22 | 11 | 100 | 0.54 | 26 | 15 | 136.7 | 0.57 | 18.4 | 7.4 | 67.2 | 0.54 |
| | d02 | 11 | 11.1 | 0.1 | 0.8 | 0.7 | 12.4 | 1.4 | 12.5 | 0.65 | 12.9 | 2 | 17.9 | 0.68 | 9.6 | -1.4 | -12.9 | 0.66 |
| | d03 | 11 | 10.3 | -0.7 | -6.2 | 0.7 | 12.2 | 1.2 | 11.1 | 0.62 | 12.2 | 1.2 | 11 | 0.67 | 9 | -2 | -18.2 | 0.64 |
| glie | d01 | 8.7 | 12.4 | 3.7 | 42.8 | 0.44 | 13 | 4.4 | 50.6 | 0.48 | 16.3 | 7.7 | 88.4 | 0.38 | 9.2 | 0.6 | 6.7 | 0.44 |
| | d02 | 8.7 | 15.4 | 6.7 | 77.3 | 0.5 | 15.4 | 6.7 | 77.5 | 0.52 | 17.8 | 9.1 | 105.2 | 0.43 | 8.9 | 0.2 | 2.4 | 0.53 |
| | d03 | 8.7 | 13.4 | 4.8 | 54.9 | 0.49 | 13.6 | 4.9 | 56.7 | 0.51 | 15.7 | 7 | 80.9 | 0.42 | 8.6 | 0 | -0.4 | 0.57 |
| amst | d01 | 26.6 | 20.2 | -6.3 | -23.8 | 0.67 | 22 | -4.6 | -17.3 | 0.62 | 26 | -0.6 | -2.1 | 0.68 | 18.4 | -8.2 | -30.9 | 0.63 |
| | d02 | 26.6 | 24.9 | -1.7 | -6.4 | 0.64 | 27.3 | 0.8 | 2.8 | 0.6 | 29.9 | 3.3 | 12.5 | 0.63 | 26.9 | 0.3 | 1.1 | 0.6 |
| | d03 | 26.6 | 23.5 | -3 | -11.4 | 0.61 | 26.5 | -0.1 | -0.3 | 0.59 | 29.5 | 3 | 11.1 | 0.59 | 29 | 2.4 | 9.2 | 0.58 |
| belz | d01 | 23.4 | 21.8 | -1.6 | -6.9 | 0.49 | 23.3 | -0.1 | -0.6 | 0.4 | 27.4 | 4 | 16.9 | 0.52 | 20.5 | -2.9 | -12.5 | 0.46 |
| | d02 | 23.4 | 22.2 | -1.2 | -5.2 | 0.45 | 24 | 0.5 | 2.3 | 0.4 | 25.9 | 2.5 | 10.7 | 0.48 | 20.3 | -3.1 | -13.2 | 0.3 |
| | d03 | 23.4 | 20.9 | -2.6 | -11 | 0.45 | 22.5 | -0.9 | -4 | 0.46 | 24.9 | 1.5 | 6.5 | 0.53 | 19.7 | -3.7 | -15.8 | 0.31 |
| brue | d01 | 28.5 | 21.8 | -6.7 | -23.6 | 0.44 | 23.3 | -5.2 | -18.3 | 0.35 | 27.4 | -1.1 | -4 | 0.45 | 20.5 | -8 | -28.2 | 0.41 |
| | d02 | 28.5 | 26.3 | -2.2 | -7.7 | 0.56 | 29 | 0.4 | 1.5 | 0.49 | 29.2 | 0.7 | 2.3 | 0.56 | 30 | 1.5 | 5.2 | 0.49 |
| | d03 | 28.5 | 24.4 | -4.1 | -14.5 | 0.56 | 27.1 | -1.4 | -5 | 0.52 | 28.3 | -0.3 | -0.9 | 0.56 | 54.2 | 25.7 | 90 | 0.48 |
| nans | d01 | 25.3 | 21.8 | -3.5 | -13.7 | 0.46 | 23.3 | -2 | -7.9 | 0.42 | 27.4 | 2.1 | 8.3 | 0.51 | 20.5 | -4.8 | -18.9 | 0.47 |
| | d02 | 25.3 | 26.3 | 1.1 | 4.2 | 0.54 | 29 | 3.7 | 14.5 | 0.52 | 29.2 | 3.9 | 15.5 | 0.6 | 30 | 4.7 | 18.7 | 0.5 |
| | d03 | 25.3 | 23.1 | -2.2 | -8.7 | 0.51 | 25.6 | 0.4 | 1.4 | 0.5 | 26.9 | 1.6 | 6.5 | 0.58 | 23.2 | -2.1 | -8.2 | 0.38 |
| pots | d01 | 15.7 | 12.4 | -3.4 | -21.5 | 0.44 | 13 | -2.7 | -17.1 | 0.33 | 16.3 | 0.6 | 3.7 | 0.35 | 9.2 | -6.5 | -41.3 | 0.31 |
| | d02 | 15.7 | 10 | -5.7 | -36.5 | 0.42 | 10.1 | -5.6 | -35.8 | 0.3 | 11.3 | -4.4 | -28.2 | 0.37 | 8.6 | -7.1 | -45.1 | 0.36 |
| | d03 | 15.7 | 9.1 | -6.7 | -42.5 | 0.4 | 9.3 | -6.4 | -41 | 0.3 | 10 | -5.7 | -36.3 | 0.35 | 7.9 | -7.9 | -49.9 | 0.36 |



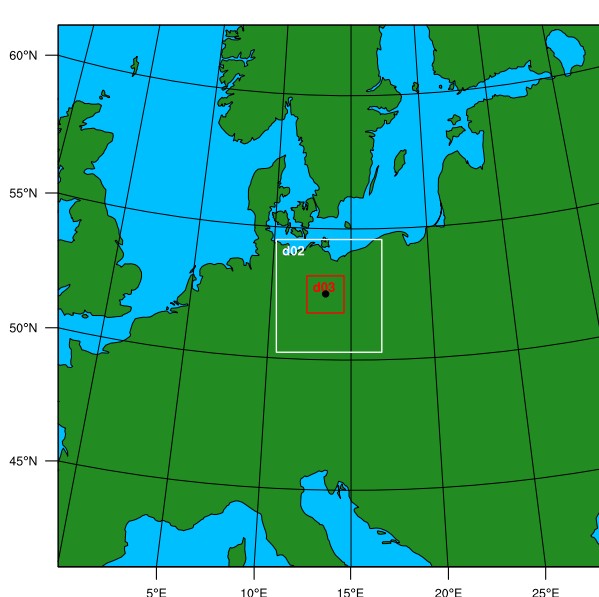

**Figure 1.** WRF-Chem model domains with horizontal resolutions of 15km (d01, outer domain), 3km (d02, middle domain) and 1km (d03, inner domain), centered around Berlin, Germany, which is marked black in the figure.





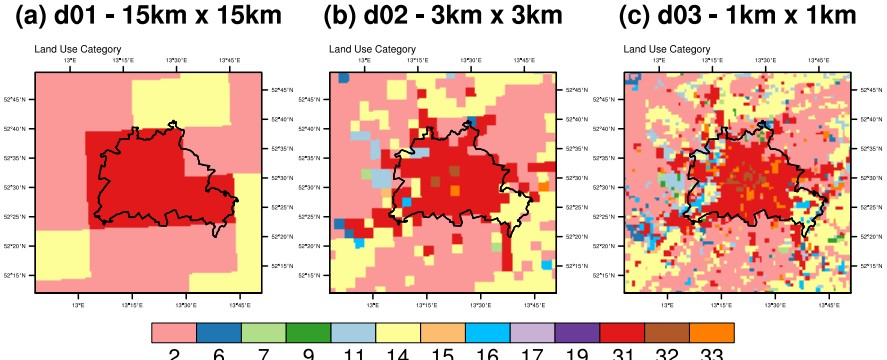

**Figure 2.** CORINE land use classes over Berlin mapped to USGS classes and interpolated to the WRF-Chem grids of (a) 15km, (b) 3km and (c) 1km horizontal resolutions. The classes are the following: 2 - dryland cropland and pasture, 6 - cropland/woodland mosaic, 7 - grassland, 9 - mixed shrubland/grassland, 11 - deciduous broadleaf forest, 14 - evergreen needle leaf forest, 15 - mixed forest, 16 - water bodies, 17 - herbaceous wetland, 19 - barren or sparsely vegetated, 31 - low intensity residential, 32 - high intensity residential, 33 - commercial/industry/transport.





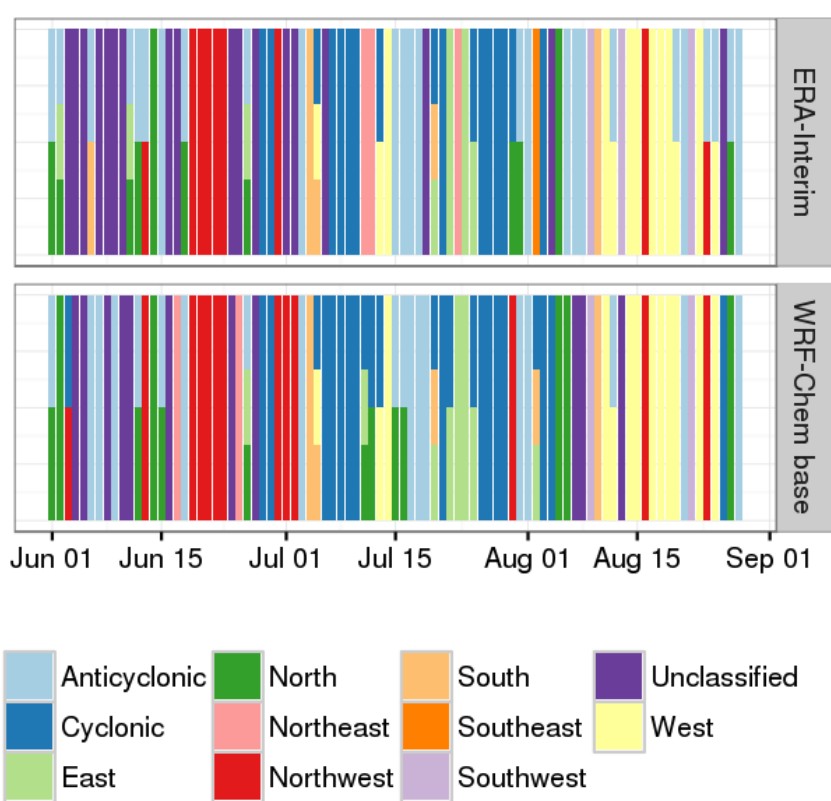

**Figure 3.** Comparison of weather types for Berlin calculated from ERA-Interim reanalysis data (top panel) and from WRF-Chem output from the domain with 15km horizontal resolution (bottom panel). Up to three weather types are calculated for each day.




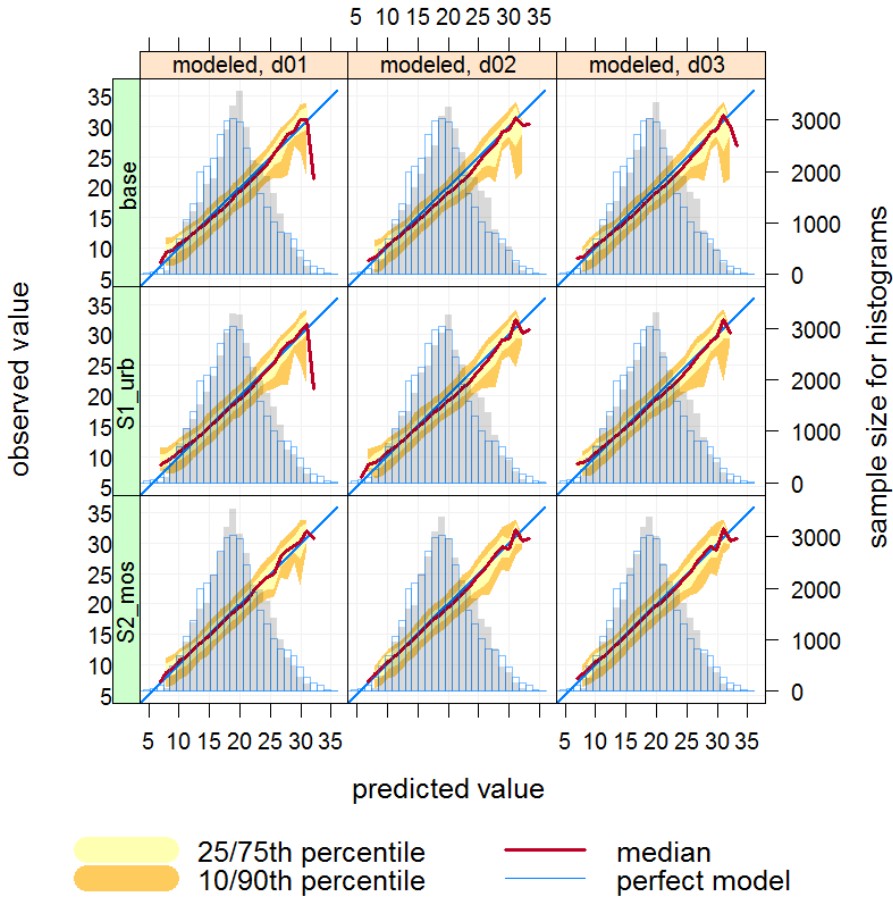

**Figure 4.** Conditional quantile plot of simulated and observed temperature (°C). The model results are split into evenly spaced bins and compared to observations spatially and temporally matching the values in the model result bins. The red line denotes the median of each of these bins. Grey bars show the distribution of model results, blue outline bars the distribution of observations. The results are shown for the base run and sensitivity simulations S1_urb and S2_mos, each for all three model domains (d01 - 15km horizontal resolution, d02 - 3km, d03 - 1km).





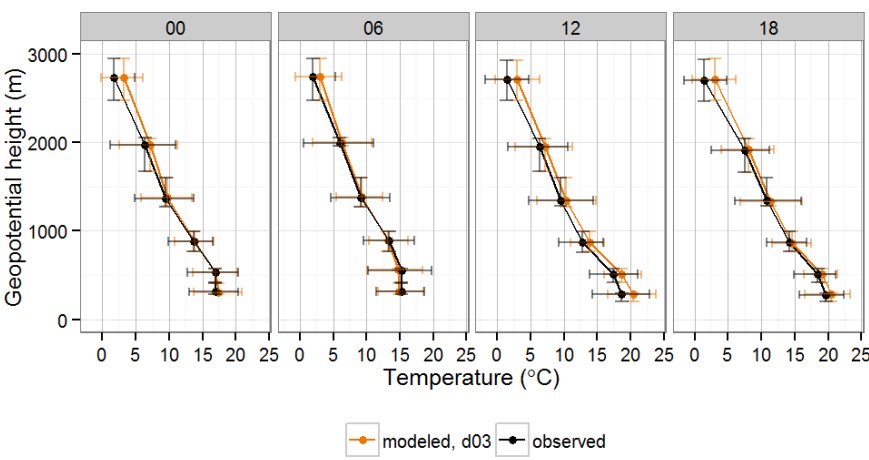

**Figure 5.** JJA mean profiles of observed and modeled (base run, 1kmx1km horizontal resolution) temperature at Lindenberg at 00:00, 06:00, 12:00 and 18:00 UTC. Error bars show the 25th and 75th percentiles of temperature and geopotential height.





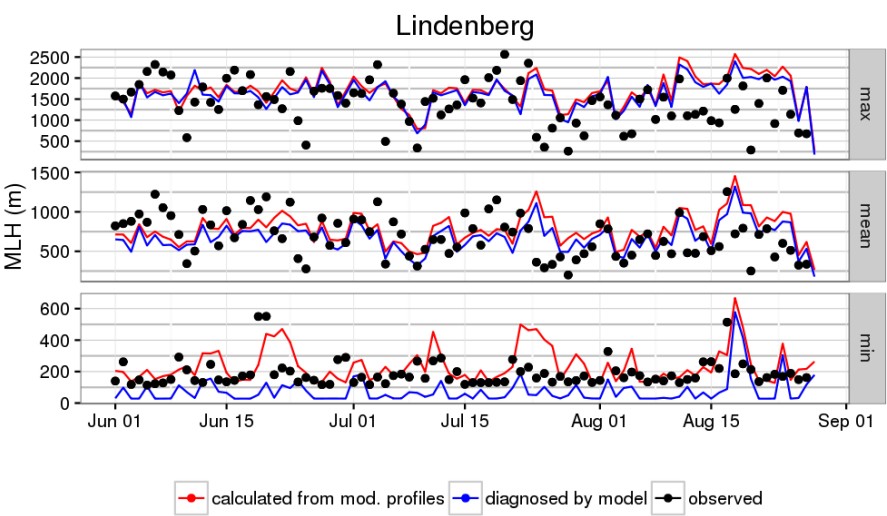

**Figure 6.** Daily minimum, mean and maximum mixing layer height as observed in Lindenberg, diagnosed by WRF-Chem and calculated from modeled profiles of temperature, wind speed and humidity (base run, 1kmx1km horizontal resolution).





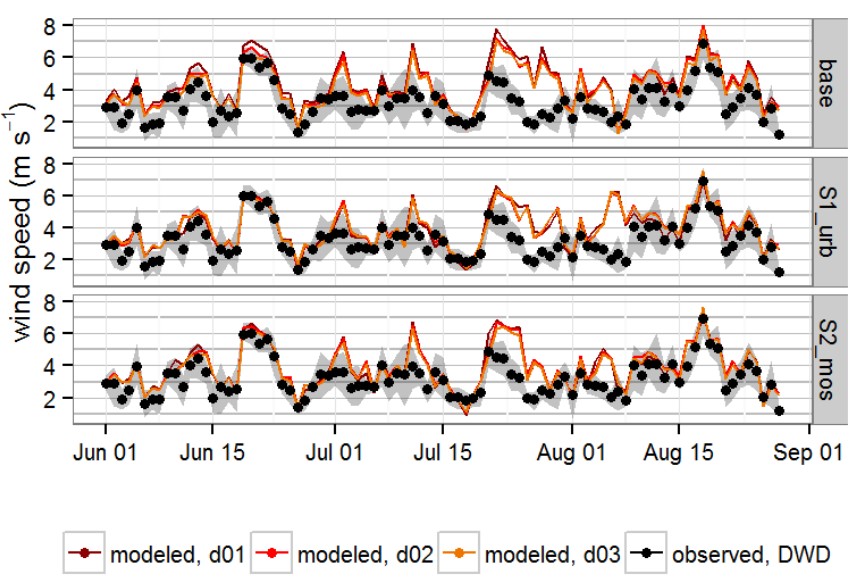

**Figure 7.** Daily mean observed and modeled wind speed from the base run, S1_urb and S2_mos, for all three model domains (d01 - 15km horizontal resolution, d02 - 3km, d03 - 1km). The figures show means over the daily means of three stations in Berlin (Tegel, Schönefeld and Tempelhof). The grey shades show the variability between the daily means of these stations, corresponding to the 25th and 75th percentiles of the individual stations' daily means. For the model results, the grid cells corresponding to the location of the stations were extracted.





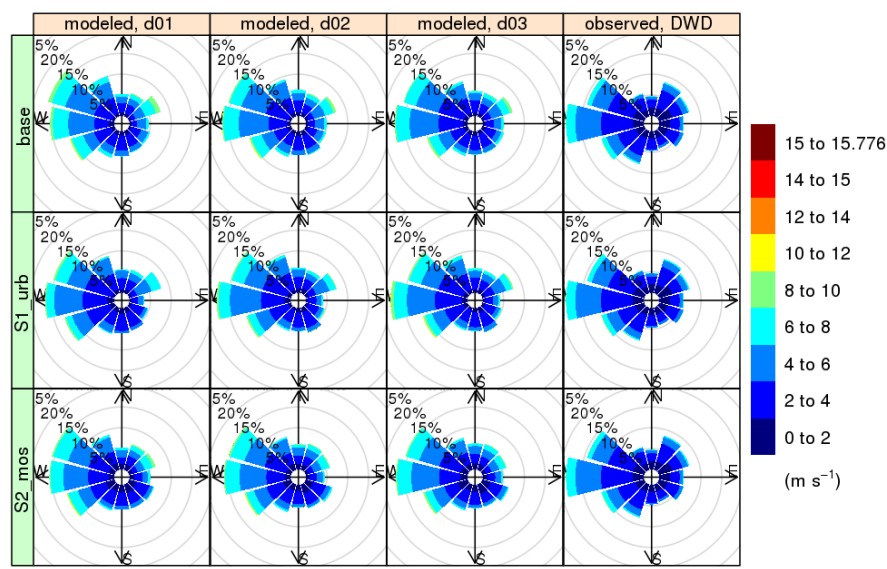

**Figure 8.** Wind roses over observed and modeled values for JJA, including observations and model results for three stations in Berlin (Tegel, Schönefeld and Tempelhof) and from all three model domains (d01 - 15km horizontal resolution, d02 - 3km, d03 - 1km). The bars refer to the frequency of how often wind was coming from the respective direction and the colors indicate how often the wind speed was observed or modeled in the indicated interval.





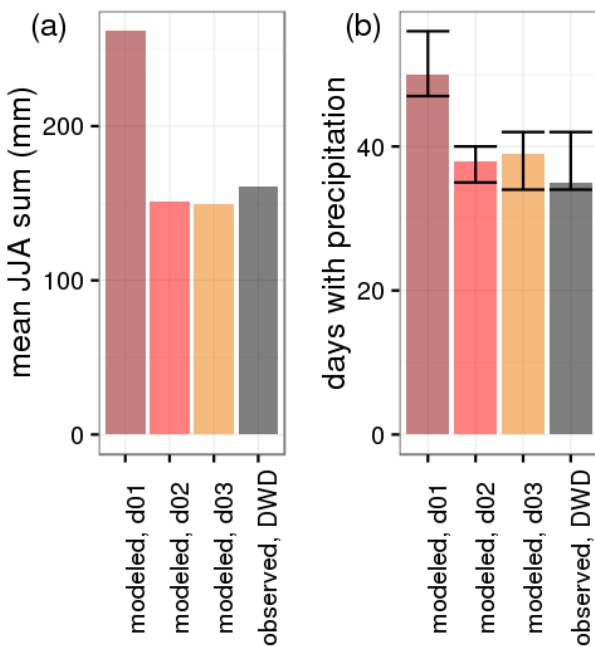

**Figure 9.** a) Station average (mean) precipitation sum of observations and model results (base run), b) median number of days with precipitation observed or modeled. A day is counted if observed or modeled precipitation was more than 1 mm/h. Ranges indicate the variability between the different stations. Both a) and b) show averages over nine stations and the corresponding model grid cells in Berlin and surroundings. Model results are given for all three model domains (d01 - 15km horizontal resolution, d02 - 3km, d03 - 1km).





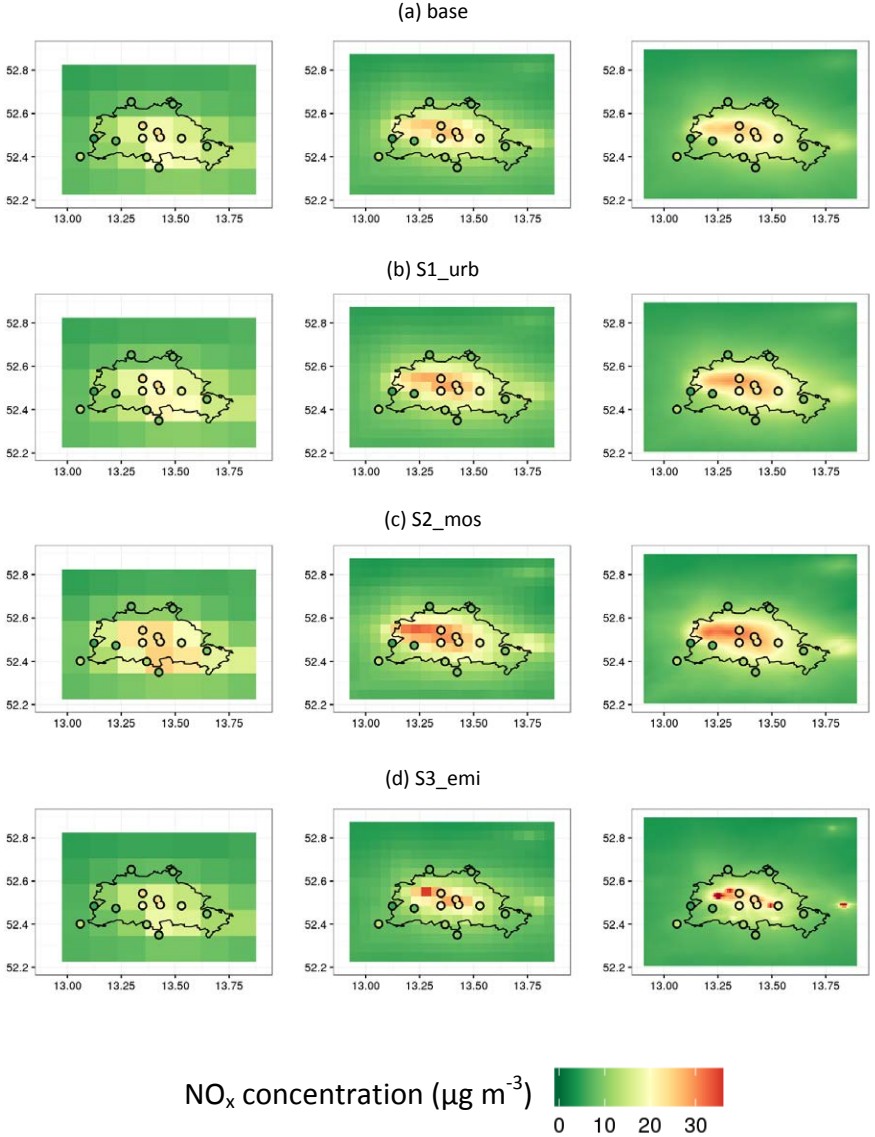

**Figure 10.** JJA mean modeled (colored fields) and observed (colored circles) $NO_x$ concentration in Berlin and surroundings from a) the base run, b) S1_urb, c) S2_mos, d) S3_emi. The left column shows results obtained with the 15km horizontal resolution, the middle results from a 3km horizontal resolution and the right column shows results from a 1km horizontal resolution.





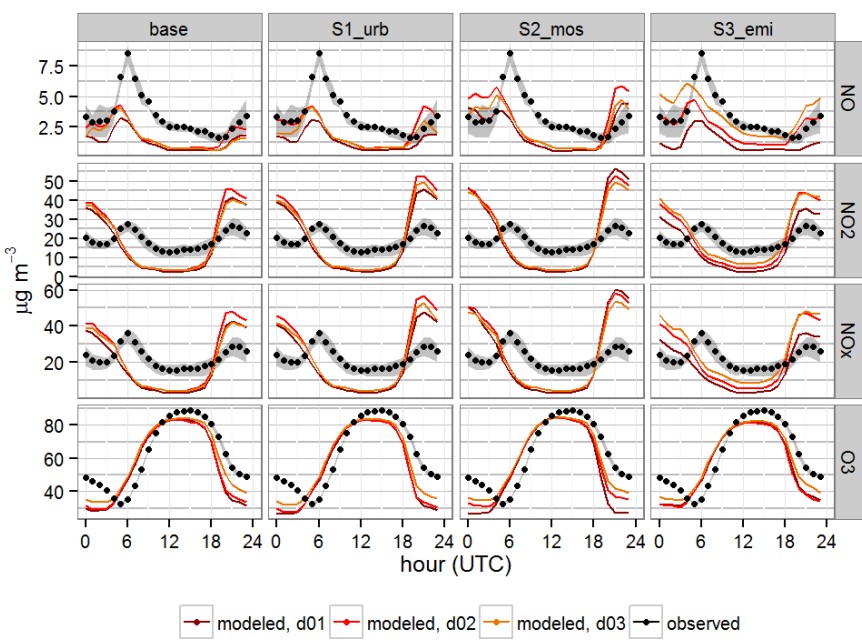

**Figure 11.** Mean diurnal cycles of NO, $NO_2$, $NO_x$ and $O_3$ for all Berlin and Potsdam urban background stations as observed and modeled by the base run, S1_urb, S2_mos and S3_emi. Model results are given for all three model domains (d01 - 15km horizontal resolution, d02 - 3km, d03 - 1km). The diurnal cycle is averaged over 6 stations for NO, $NO_2$ and $NO_x$ and 3 stations of $O_3$. The grey shaded areas represent the variability between the different stations' diurnal cycles, showing 25th and 75th percentiles.



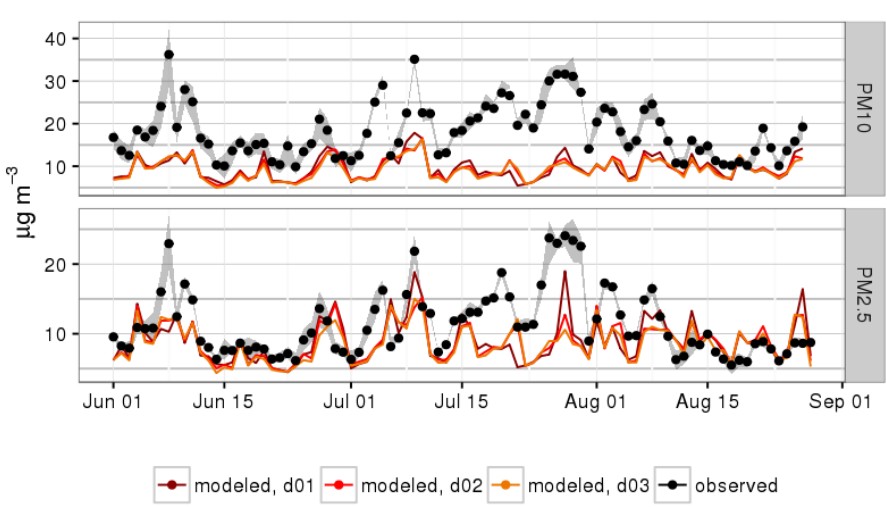

**Figure 12.** Daily mean PM$_{10}$ and PM$_{2.5}$ concentrations as observed and modeled (base run) at urban background stations in Berlin. Daily means are averaged over five stations for PM$_{10}$ and four stations for PM$_{2.5}$. The grey shaded areas represent the variability between the different stations, showing 25th and 75th percentiles. Model results are given for all three model domains (d01 - 15km horizontal resolution, d02 - 3km, d03 - 1km).





**Figure 13.** Differences in nighttime (20:00 - 02:00 UTC) mean JJA planetary boundary layer height as diagnosed from WRF-Chem, a) S1_urb - base run, b) S2_mos - base run (at 1km horizontal resolution).