# Peer review of "Air quality modelling in the Berlin-Brandenburg region using WRF-Chem v3.7.1: sensitivity to resolution of model grid and input data"

_Geoscientific Model Development, 2016_

## Referee Comment (RC1) · Anonymous Referee #1 · 1 Sep 2016

**General vote**

With regard to increasing urbanisation and connected air pollution, improving air quality models in urban areas gets more and more important in order to understand the complex interactions between physical and chemical processes in these highly dynamic environments.

The existing paper nicely presents a regional WRF-Chem modelling study for the urban area of Berlin discussing the model sensitivity to varying resolution and different input data. It is well structured and written in an understandable way, which makes it easy to follow and get the aim of the work. The relevance for the scientific field is given on the one hand by using a novel methodology coupling a modified version of the WRF Urban Canopy Model to WRF-Chem and further relating to existing relevant studies in the field.

Before being suitable for publication in 'Geoscientific Model Development' however some important points have to be clarified and discussed further, which are mentioned in the following:

**General concern**

Could you please justify the use of the single layer urban canopy model (SLUCM) instead of the more complex multi layer approach BEP. If I am not wrong, the BEP allows for a higher vertical resolution to the ground and therefore might be more suitable to discuss 'ground level concentrations' in urban areas. It would be nice at one point to give more details on the vertical resolution. Using the SLUCM would mean that your lowest model level would be located at the level of the mean building height defined in the parameter table, which could lead to significant discrepancies when comparing simulated concentrations which local air quality observations. I guess the underestimation of PM10, PM2.5 and NO might be partly to the fact, that the lowest model level is too far away from the emission source/measurement height, so that a certain amount of pollutants already have been dispersed by vertical mixing.

In my point of view, this aspect has to be mentioned and discussed somewhere in the paper.

Minor review points are listed below on a line-by-line basis:

Page 1:

13**:** specify the characteristics of the 'base run'

14: more information on the term 'mosaic option' needed. The benefit/aim of using this approach is not really clear.

Page2:

28: What is the expected benefit in using the SLUCM? Would BEP also be an option as well. Please discuss this aspect in the proceeding of the study.

Page 3: 'Among[...]' : please check language

6-9: At this point you mention the multi-layer model BEP used by Fallmann et al. (2016). Please discuss, why you have chosen the SLUCM for your study.

29:  Please provide information on your vertical resolution.

1: What do you mean with 'default input parameters'? US-City? Where are they originated from?

7: How did you calculate the mean building length in QGIS? Did you use all the buildings within the city's vicinity? Was there an algorithm? Please describe briefly.

24: How is the distributions improved around cities? Is there literature on that?

26: height of the first model layer?

3: Indicate briefly why a second quality control is needed and how it is achieved.

6: better: 'specific humidity data'

7-15: Was the calibration and quality check of the sensors part of the study? What is meant by visual inspection? Maybe you can leave out 11-14..

17: The Lindenberg station is located in some distance to the urban area of Berlin. Please discuss why the profile data is suitable for your approach as the SLUCM only works for urban areas and modifies the local meteorology there. Additionally, the Lindenberg profiles start in a height which is located over the mean heights of the buildings I guess. With regard to Fig. 5, especially during noon, the model overestimates the observed temperature by 2-3 degrees. Please provide some discussion on that in later chapters (Chapter 3.2.2)

 1-17:

What is the model reference height for comparisons? For primary pollutants in particular, there is a rapid decrease in concentration with height due to mixing, deposition and chemical reactions. This effect in addition to the fact that you are comparing a grid cell to a point measurement might be the predominant reason for an underestimation. Please provide further discussions on that point in later chapters.

23: The simulated 2m temperature is not the actual temperature, but a diagnostic variable in your WRF-Chem setup, correct?

2: The negative bias in that study could also be related to the vertical resolution.

3: see above

8-10: Please describe more into detail.

14: 2m temperature?

21ff: The general ability of your WRF setup to reproduce the meteorology of a 'non-urban' station does not reveal a lot about the models skill in the urban area of Berlin. Please see previous comment for Page10 Lines 1-17

Page 12:

4: is it possible to show the ceilometers observations?

6: On which basis is the MLH calculated in YSU?

 Page 13:

19: '..the chemical mechanism itself..'

29: Vertical resolution could be main source of error.

33: better: tends to be most pronounced...

34: why?

Page 14:

13: Can you see a relation between NOx and O3 with regard to photochemical reactions?

Page15:

 2: see above (vertical resolution)

10: '...and resolution'

17: wind speed in which height?

30: better: 'bias is reduced'

5ff: Please discuss at that point the effect of vertical and horizontal resolution as well.

22: mention vertical resolution

Page 18:

24: Specify the height of the first model level.

6 Summary and Conclusion

As mentioned in previous points, the only point of concern which is consistent throughout the study is the vertical resolution and the connected errors in reproducing the pollutant concentration close to the ground. For a sensitivity study looking at differences between similar model configurations this might not be much of a problem, but at least has to be discussed more detailed. I was wondering if 'better' results would have been achieved when using BEP instead. I know that the mosaic approach does not exist for the BEP. It would be interesting to quantify the effect of using a modified SLUCM (mosaic) in comparison to BEP to get a feeling of the sensitivity towards the urban canopy scheme.

---

## Referee Comment (RC2) · Anonymous Referee #2 · 7 Oct 2016

In this study the authors applied the WRF-Chem model with various configurations to simulate air quality in the Berlin-Brandenburg region. The impact of the grid resolution, the urban canopy parameterization in WRF and the spatial resolution of the anthropogenic emission inventory on the model performance were studied. It is crucial to develop and evaluate the state of the art air quality modeling tools for large urban areas, where complex meteorology and anthropogenic emissions make it harder to accurately predict air pollution. The authors used a large amount of measurement data to evaluate the simulated meteorological and chemistry variables. I think the paper deserves publication in GMD after addressing the following comments:

You used the RADM2 mechanism. This mechanism doesn't include several important biogenic VOCs, therefore it usually underestimates ozone compared to such gas chemistry mechanisms as RACM. I suggest this point to be discussed in the paper.

You used 35 vertical levels, how thick is the first layer? You did simulations as high as 1km resolution, which certainly helps to capture spatial variability of the urban canopy and anthropogenic emissions in more detail. However, the vertical resolution remained the same. I think relatively coarse vertical resolution could explain some of the model deficiencies, especially the nighttime bias in the model.

Chapter 2.1- doesn't the ERA-INTERIM have 61 vertical levels?

2.2- WRF also uses more updated MODIS LU dataset. Here the USGS LU data is mentioned only.

2.4- The stack height could be small, but the plume injection height due to buoyancy and momentum is higher. I think this could explain some of the NOx overestimations by the model at nighttime, when all the NOx from the point sources are emitted into shallow boundary layers.

In WRF-Chem the vertical mixing of the chemical species are done somewhat differently than the meteorological fields. Did you consider testing sensitivity of the vertical mixing of the chemical species to various parameters/assumptions? The treatment of vertical mixing of the chemical species can explain some of the high NOx bias at night.

In addition, I suggest showing vertical profiles or x-sections of the chemical species (e.g. NOx) to illustrate how deep the species were mixed in the model, especially during nighttime.

Can you show model-obs. comparisons similar to Figure 11 for NOy (if measurements are available) as well? The paper doesn't show any evaluations for CO, biogenic VOCs such as isoprene etc.

---

## Author Comment (AC1) · 3 Nov 2016

**Authors' reply to comments of Referee 1**

We thank Referee 1 for the very helpful feedback on the manuscript. In the following, we address the comments, listing the Referee's comment (bold), our response (normal font) and changes in the manuscript, if applicable (indented). All references given in this response can be found in the bibliography of the discussion paper or below each response. In addition to the changes replying to the Referees' comments, we made few further minor changes to the manuscript, listed below.

**RC1 – General concern**
**Could you please justify the use of the single layer urban canopy model (SLUCM) instead of the more complex multi layer approach BEP. If I am not wrong, the BEP allows for a higher vertical resolution to the ground and therefore might be more suitable to discuss 'ground level concentrations' in urban areas. It would be nice at one point to give more details on the vertical resolution. Using the SLUCM would mean that your lowest model level would be located at the level of the mean building height defined in the parameter table, which could lead to significant discrepancies when comparing simulated concentrations which local air quality observations. I guess the underestimation of PM10, PM2.5 and NO might be partly to the fact, that the lowest model level is too far away from the emission source/measurement height, so that a certain amount of pollutants already have been dispersed by vertical mixing. In my point of view, this aspect has to be mentioned and discussed somewhere in the paper.**
Thank you for your comments. The Building Effect Parametrization (BEP) urban canopy model allows for a more detailed parametrization of meteorological processes within the urban area and includes the possibility of considering buildings extending through several model layers (see, e.g., Skamarock et al., 2008; Chen et al., 2011). It does however not add an additional parametrization of the diffusion of chemical species within street canyons. When using the BEP it is recommended to increase the vertical resolution in the urban canopy (Chen et al., 2011). Using the BEP and increasing the vertical resolution accordingly would increase even further the computational cost, which is already very high when running WRF-Chem at a horizontal resolution of 1km. In addition, it requires a much more detailed specification of the urban structure as input to the model. Limitations in combining the BEP with the WRF-Chem setup we used are its incompatibility with the mosaic option and the restriction on the coupling with planetary boundary layer schemes from both sides. There is only one PBL scheme suitable to be combined with both WRF-Chem and the BEP (MYJ), which is not the PBL scheme chosen for this study.

Jänicke et al. (2016) tested different combinations of urban canopy models and planetary boundary layer (PBL) schemes and concluded that using the BEP did not outperform other approaches concerning 2m air temperature. They also tested modifying the vertical model resolution (40 levels instead of 28 levels as their regular set-up), which did not lead to considerable improvements (not more than ±0.1 K) in the bias or root mean square deviation of simulated 2m temperature in urban areas for a set-up with BEP in combination with the Bougeault-Lacarrére PBL scheme at 2 km horizontal resolution. Furthermore, Mar et al. (2016) tested the sensitivity of simulated air pollutant concentration to vertical resolution and did not see a big difference using a higher number of vertical levels.

We agree that the bias in simulated air pollutant concentrations could partly be due to limits in the comparability between grid-cell averaged simulated concentrations and observations at single locations, with air pollutants typically sampled at altitudes of 2.5m-3m above the surface. However, since it is the volume of a grid box that determines the initial dilution of an air pollutant after emission, we expect the horizontal resolution of the model (and of the emission data) to play a more important role than the vertical resolution. In our case, the emission data have a native resolution of 7x7 km and even after downscaling the data to the model grid of 1x1 km, this is rather coarse for highly heterogeneous domains such as a city. This is a basic problem of modeling air pollution with any model at this kind of resolution.

We agree that it would be interesting to investigate, in future studies, the impact of a higher vertical resolution as well as of using the BEP on simulated air pollutant concentrations. Following your suggestions, we included a discussion of all of these issues and in particular of the horizontal and vertical resolution issues for the model evaluation in different sections of the manuscript as well as in the conclusions, as detailed in our responses to your specific comments below.

**Page 1:**
**13:**
**specify the characteristics of the 'base run'**
The abstract introduces the general settings of the base run in lines 6-8. For clarity, we refer again to those characteristics as follows.

> "The results show that the model simulates meteorology well, though urban 2m temperature and urban wind speeds are biased high and nighttime mixing layer height is biased low in the base run with the settings described above."

**14:**
**more information on the term 'mosaic option' needed. The benefit/aim of using this approach is not really clear.**
The "mosaic option" is used to allow for a sub-grid scale specification of land use. As it is introduced with this term in lines 9-10 of the abstract, we prefer not to make any changes to the abstract at this point.

**Page2:**
**28:**
**What is the expected benefit in using the SLUCM? Would BEP also be an option as well. Please discuss this aspect in the proceeding of the study.**
We have decided to use the SLUCM, as the present setup at a horizontal resolution of 1kmx1km is already very computationally expensive. Using the BEP would further increase the computational expense, require additional input data describing the urban structure, and not be compatible with the PBL scheme and the mosaic option used here. As for simulating urban temperatures, Jänicke et al. (2016) found that the more complex urban canopy models did not perform better than a simple bulk approach at 2kmx2km resolution. We specify this in section 2.3 as follows. Please also see our reply to your general comments, as well as reply and changes in the manuscript regarding your comments on the conclusions.

> Page 5, line 29-33: "We use the single-layer urban canopy model […]. We choose to not use a more complex parametrization of the urban canopy, such as the building effect parametrization (BEP), because the computational cost is already very high at a horizontal resolution of 1kmx1km, and a more complex parametrization of the urban canopy, along with the required increase of vertical model resolution, would increase the computational cost further and require a more detailed input dataset describing the urban structure. Moreover, the BEP is not applicable with the mosaic option in WRF so far and the only applicable PBL scheme in combination with the BEP and WRF-Chem is the Mellor-Yamada-Janjic scheme. This scheme often lead to stronger biases in simulated 2m air temperature than other parametrizations such as the YSU scheme (Hu et al., 2010; Loridan et al., 2013; Jänicke et al., 2016), the scheme selected for this study. In addition, Jänicke et al. (2016) could show that the BEP did not outperform simpler approaches such as the bulk scheme or the single-layer urban canopy model with respect to simulating 2m temperature and that the PBL scheme had stronger influence on simulated 2m air temperature than the urban canopy parametrization."

> Hu, X.-M., J. W. Nielsen-Gammon and F. Zhang (2010): Evaluation of three planetary boundary layer schemes in the WRF model. J. Appl. Meteor. Climatol. 49 (9): 1831-1844. doi: 10.1175/2010JAMC2432.1

Loridan, T., F. Lindberg, O. Jorba, S. Kotthaus, S. Grossman-Clarke and C. Grimmond (2013): High Resolution Simulation of the Variability of Surface Energy Balance Fluxes Across Central London with Urban Zones for Energy Partitioning. Bound.-Layer Meteor. 147 (3): 493-523. doi: 10.1007/s10546-013-9797-y

**Page 3, line 13:**
**'Among[...]' : please check language**
We checked the language and will stick to our initial formulation.

**Page 4:**
**6-9:**
**At this point you mention the multi-layer model BEP used by Fallmann et al. (2016). Please discuss, why you have chosen the SLUCM for your study.**
Please see our reply and changes in the manuscript concerning page 2, line 28.

**29:**
**Please provide information on your vertical resolution.**
The first layer is at ca. 30m above the surface. We will add this information to the model description.
> P. 4, line 28-29: "The model top is at 50 hPa, using 35 vertical levels. The first model layer is at approximately 30m above the surface, with 12 levels in the first 3km. "

**Page 6:**
**1:**
**What do you mean with 'default input parameters'? US-City? Where are they originated from?**
The default input parameters are specified in a file read in by the WRF-Model. It was developed in the US for typical American urban conditions, but we cannot say with certainty to which city they correspond. The original file including the default settings (downloadable with the source code) recommends the user to adapt the parameters to the case of interest. We make this clearer in the manuscript.
> "In our base simulation, we use the default input parameters as specified in the look-up table included in the standard distribution of the WRF source code available from UCAR."

**7:**
**How did you calculate the mean building length in QGIS? Did you use all the buildings within the city's vicinity? Was there an algorithm? Please describe briefly.**
The software allows the user to calculate the area of each building-geometry, and the mean building length was calculated by taking the square root of the area. The dataset only included buildings within Berlin, so only those were used as a basis for this calculation. We clarified this in the manuscript as follows.
> Page 6, line 3-4: "The calculations are based on detailed maps of Berlin provided by the […]"
> Page 6, line 6-7: "[…]and the mean building length is calculated with the software QGIS, by calculating the surface area of each building geometry in the dataset and assuming its square root as each building's mean length. "

**24:**
**How is the distributions improved around cities? Is there literature on that?**
In the H2020 project MACC-III the time-series of the TNO-MACC-II 2003-2009 inventory published by Kuenen et al. (2014) was substantially expanded to cover all years between 2000 and 2011. A more recent version of the official reported data (reported as of 2013) was included causing changes in the national total emissions compared to the TNO-MACC-II version. Moreover, the spatial distribution of emissions was improved, providing a better representation of major cities in the emission grids. The latter was especially improved by no longer using population density as a default for diffuse (nonpoint source) industrial emissions but using industrial land use as a distribution proxy. Residential solid fuel use (wood, coal) was on a per capita basis allocated more in the rural areas than large city centers. Other improvements but less relevant for the present paper included a new set of shipping emissions data, which was also adopted by EMEP, and the inclusion of CO2 emissions including a split between fossil fuel CO2 and CO2 from biofuels. We include more detailed information on this in the manuscript as follows.

> Page 6, line 22: "In comparison to version II of the inventory (Kuenen et al., 2014), version III includes, amongst other updates, an improved distribution of emissions especially around cities. The distribution was improved by no longer using population density as a default for diffuse (non-point source) industrial emissions but using industrial land use as a distribution proxy. Residential solid fuel use (wood, coal) was allocated more to rural areas than to large city centers, on a per capita basis."

**26:**
**height of the first model layer?**
Please see our reply to your comment and changes in the manuscript concerning page 4, line 29.

**Page 8:**
**3:**
**Indicate briefly why a second quality control is needed and how it is achieved.**
The second quality control is also done by the German Weather Service. Since the wording in the manuscript is somewhat ambiguous, we change it as follows:

> "A second-level quality control, as described in Kaspar et al. (2013), has been applied to the data."

**6:**
**better: 'specific humidity data'**
We adapted the changes as you suggested.

> "In addition, we use specific humidity data from […]"

**7-15:**
**Was the calibration and quality check of the sensors part of the study? What is meant by visual inspection? Maybe you can leave out 11-14..**
The calibration and quality check was not part of this study. As you suggest, we leave out lines 11-14 and rephrase the paragraph as follows.

> "The Chair of Climatology of Technische Universität Berlin (TU) runs an urban climate observation network (Fenner et al., 2014), from which we use observations of 2m air temperature to complement observations from DWD stations. We include this additional data source, as many of the TU stations are situated in urban built-up areas (see Table 3). We use quality-checked data aggregated to hourly mean values."

**17:**
**The Lindenberg station is located in some distance to the urban area of Berlin. Please discuss why the profile data is suitable for your approach as the SLUCM only works for urban areas and modifies the local meteorology there. Additionally, the Lindenberg profiles start in a height which is located over the mean heights of the buildings I guess. With regard to Fig. 5, especially during noon, the model overestimates the observed temperature by 2-3 degrees. Please provide some discussion on that in later chapters (Chapter 3.2.2)**
Your comment pointed us to the fact that one measurement/model point was missing in Fig. 5, which has been updated as inserted below. The radiosonde observations do indeed start above the height of the buildings in Berlin. You are also right that the Lindenberg site is located outside of Berlin within a rural area. As we have very little observational data available to compare the simulated

vertical structure of the atmosphere with observations, we included this comparison in the manuscript. This does not particularly target the evaluation of the effect of the SLUCM, but rather the model performance in simulating the vertical structure in the lowest 2-3 km in general, which is mainly determined by the large-scale meteorology.  The difference between model and observations at the lowest point is below 1°C, which corresponds to what is written in the manuscript (p. 11, line 22). We added the following to the manuscript.

> Page 10, line 13: "Even though observations of temperature profiles are only available outside of the urban area of Berlin, we include this comparison, in order to get a general impression of how the model performs in simulating the vertical atmospheric structure in the lowest 2-3 km."

[Figure]

**Page 10:**
**1-17:**
**What is the model reference height for comparisons? For primary pollutants in particular, there is a rapid decrease in concentration with height due to mixing, deposition and chemical reactions. This effect in addition to the fact that you are comparing a grid cell to a point measurement might be the predominant reason for an underestimation. Please provide further discussions on that point in later chapters.**
Thank you for your comment. The reference height is the height of the first model layer. We will include the following discussion:

> Page 13, line 22: "A further reason for the model bias might also be the principal challenge of comparing grid-cell averages with point observations, particularly in regions with a high variability on small spatial scales which is quite typical for cities.  Regarding the relatively coarse vertical resolution of the model, extrapolation from the first model level to the surface (e.g. Simpson et al., 2012) might allow for a better comparability between model and observation. The spatial representativeness of a measurement site for a larger area such as the 1x1 km grid cells, however, might be somewhat limited particularly for urban background sites, which can be influenced by local sources and subgrid-scale variations in emissions that cannot be captured with WRF-Chem."

Simpson, D., Benedictow, A., Berge, H., Bergström, R., Emberson, L. D., Fagerli, H., Flechard, C. R., Hayman, G. D., Gauss, M., Jonson, J. E., Jenkin, M. E., Nyíri, A., Richter, C., Semeena, V. S., Tsyro, S.,

Tuovinen, J.-P., Valdebenito, Á., and Wind, P.: The EMEP MSC-W chemical transport model – technical description, Atmos. Chem. Phys., 12, 7825-7865, doi:10.5194/acp-12-7825-2012, 2012.

**23:**
**The simulated 2m temperature is not the actual temperature, but a diagnostic variable in your WRF-Chem setup, correct?**
That is correct. We specify this as follows:

> Page 9, line 9: "We then focus on evaluating the modelled meteorology including the diagnostic variables 2m temperature (T2), 10m wind speed and direction (WS10 and WD10), […]"

**Page 11:**
**2:**
**The negative bias in that study could also be related to the vertical resolution.**
We agree with the reviewer and added a discussion on this issue to the manuscript (please see our answers to "RC1 – General concern" and to "Page 10: 1-17" above).

**3:**
**see above**
Please see our response to your comment concerning page 11, line 2, as well as our discussion of the temperature bias following page 11, line 2.

**8-10:**
**Please describe more into detail.**
This comment refers to the description of the bias of daily maximum temperature and its change with horizontal resolution. We would like to refer you to Tables 5 and S4 mentioned in the same paragraph, as well as the discussion following page 11, line 10.

**14:**
**2m temperature?**
Yes, please see our changes to the manuscript.

> Page 11, line 13: "Secondly, while the modeled 2m temperatures generally differ […]"

**21ff:**
**The general ability of your WRF setup to reproduce the meteorology of a 'non-urban' station does not reveal a lot about the models skill in the urban area of Berlin. Please see previous comment for Page10 Lines 1-17**
Please see our response to your comment and our changes in the manuscript concerning page 8, line 17.

**Page 12:**
**4:**
**is it possible to show the ceilometers observations?**
We included a figure showing the ceilometer observations in the supplementary material (see also below). The figure shows the comparison of modeled PBL height (diagnosed by WRF-Chem) at the Nansenstraße location with the mixing layer height derived from ceilometer observations at Nansenstraße. The results are shown for all three model resolutions (d01 – 15km, d02 – 3km, d03 – 1km). However, as we discuss in the manuscript, we do not know to which degree the results are quantitatively comparable (see also answer to next comment below).

[Figure]

**6:**
**On which basis is the MLH calculated in YSU?**
The calculation of the mixing layer height in YSU is essentially based on the Richardson number. However, the critical Richardson number used in the YSU scheme is 0. We include this in the manuscript.

> Page 10, lines 12-15: "The modeled MLH is compared to observations in two different ways: firstly, using the planetary boundary layer height directly diagnosed by WRF-Chem, which in the YSU scheme is calculated based on comparing the Richardson number with a critical value of 0 (Hong et al., 2005). Secondly, by calculating…"
> Page 12, lines 6-10: "The comparison of daily minimum MLH with ceilometer observations also shows an underestimation of MLH-YSU in the same range as at Lindenberg (figure S9 in the supplementary material). However, we do not know whether the magnitude of the mixing layer height derived from the ceilometer backscatter profile is directly comparable with the mixing layer height calculated from profiles of temperature, wind speed and humidity or with the mixing layer height calculated by the model. This makes it more difficult to evaluate the modeled mixing layer height quantitatively at the urban site Nansenstraße. For this, further studies assessing the comparability of MLH derived from radiosonde and ceilometer observations would be necessary."

Hong, S.-Y., Noh, Y. and Dudhia, J., 2006: A New Vertical Diffusion Package with an Explicit Treatment of Entrainment Processes, *Mon. Wea. Rev.,* **134**, 2318–2341, doi: 10.1175/MWR3199.1

**Page 13:**
**19:**
**'..the chemical mechanism itself..'**
We've made the changes in your script according to your suggestion.

> "[…] the chemical mechanism itself: […]"

**29:**
**Vertical resolution could be main source of error.**
Mar et al. (2016) have tested the sensitivity of the simulated air pollutant concentrations to a higher number of vertical levels, including a lower first model level, and found that this does not have a large impact on the simulated concentrations. However, we agree that future studies could investigate the issue of vertical resolution in more detail for model setups with high horizontal resolution. Please also see our response to your "general concern" and our replies, as well as changes in the manuscript, to your comment concerning page 10, lines 1-17 and the summary.

**33:**
**better: tends to be most pronounced...**
We are not sure which sentence you are referring to exactly.

**34:**

**why?**

We presume your comment points to the statement that daytime concentrations at suburban and rural sites are biased less than urban background concentrations. One possible explanation for this is given on page 14, line 1-3, directly following the paragraph you commented: we believe that the relatively coarse resolution of the emission inventory (7km) leads to an underestimation of emissions within the urban area, and along with this an underestimation of simulated concentrations in the urban areas, while the emissions are more representative in the rural or suburban background at a horizontal resolution of 7km.

**Page 14:**

**13:**

**Can you see a relation between NOx and O3 with regard to photochemical reactions?**

We have specified the relation in the manuscript as follows:

> "The mean O3 is still simulated reasonably well, though the model underestimates at night and overestimates during the morning hours. The bias is consistent with a bias in NOx diurnal cycles discussed above: in particular, the underestimation of $O_3$ during nighttime is consistent with an overestimation of NOx; the overestimation of $O_3$ in the morning hours might result from too much $NO_2$ accumulating at the surface, which is photolyzed when the sun rises."

**Page 15:**

**2:**

**see above (vertical resolution)**

Please see our response to your comment concerning page 13, line 29, our response to your comment regarding page 10, line 1-17 as well as our response to your "general concern".

> Page 14, line 30: "[…] more than PM2.5. As for the simulated chemical species, part of the bias might be due to a somewhat limited comparability of grid-cell averaged particulate matter with observations at a measurement site. It should further be noted […]"

**10:**

**'...and resolution'**

We have added this as you suggested.

> "[…] is mainly due to an underestimation of secondary organic aerosols by the aerosol mechanism as well as missing emissions, and potentially also the vertical resolution as previously discussed."

**Page 16**

**17:**

**wind speed in which height?**

The wind speed is measured at 10m and compared to the modeled 10m wind speed, which is a diagnostic variable in WRF. We mention this on page 9, line 9. However, we included it again as follows.

> "The bias in 10m wind speed is reduced […]"

**30:**

**better: 'bias is reduced'**

We followed your suggestion.

> "[…] though the bias in the diurnal cycles of T2 and wind speed is reduced more in S1_urb."

**Page 17**

**5ff:**

**Please discuss at that point the effect of vertical and horizontal resolution as well.**

Following your comment concerning page 10, lines 1-17, we have included a discussion of this previously. We have also included it in this section as follows.

> Page 17, line 23: "[…] in section 4.2. As previously mentioned, a further reason for this bias might be limitations in comparability between grid-cell averaged simulated concentrations and point observations near the surface."

**22:**

**mention vertical resolution**

Please see our response to your comment and changes in the manuscript concerning page 17, lines 5ff.

**Page 18:**
**24:**

**Specify the height of the first model level.**

Following your comment concerning page 4, line 29, we have added this information in the description of the model setup. Please see our changes in the manuscript replying to your comment concerning page 4, line 29.

**RC1 – summary**

**As mentioned in previous points, the only point of concern which is consistent throughout the study is the vertical resolution and the connected errors in reproducing the pollutant concentration close to the ground. For a sensitivity study looking at differences between similar model configurations this might not be much of a problem, but at least has to be discussed more detailed. I was wondering if 'better' results would have been achieved when using BEP instead. I know that the mosaic approach does not exist for the BEP. It would be interesting to quantify the effect of using a modified SLUCM (mosaic) in comparison to BEP to get a feeling of the sensitivity towards the urban canopy scheme.**

Thank you again for your constructive feedback. For reasons we have specified in our response to your general concern, we have not tested the BEP. However, an interesting – though computationally expensive – comparison would be the one you are suggesting. We have included this in our conclusions as follows.

> Page 19, lines 3-6: "As for the vertical distribution of emissions as well as an increased vertical model resolution, Mar et al. (2016) state it has little impact on the model results. While this might hold for simulations of rural background air quality with domain resolutions of the order of 45km, the present results suggest that it is of higher relevance to distribute point source emissions into several vertical model levels when decreasing the model resolution and the resolution of the emission input data. Similarly, increasing the vertical model resolution at the same time might both help distribute emissions better and improve the modeled mixing."

> Page 20, lines 3-11: "A more detailed specification of urban land use classes together with the respective input parameters can help better represent the heterogeneity of urban area in a model domain with 1km horizontal resolution. This is shown by the modeled 2m temperature only differing by more than 0.1°C between the model resolutions of 3km and 1km if the land use class of the respective grid cell changes. It is further shown by the simulation with updated urban parameters decreasing the positive bias in simulated wind speed in the base run by up to 0.5 m/s, from a mean bias in wind speed up to 1.5 m/s in the

base run to a mean bias in wind speed of maximally 1m/s in the sensitivity simulation where urban parameters have been updated. In addition, the nighttime mixing layer height is simulated higher in this sensitivity simulation for grid cells of the urban types high intensity residential and commercial/industry/transport, suggesting that the negative bias in mixing layer height at nighttime can also be corrected by better specifying the input parameters to the urban scheme and the urban land use classes. Further studies could target a comparison between the urban parametrization used in this study with the more complex – and computationally expensive - approach of representing the urban meteorology with the building effect parametrization (BEP) urban canopy model combined with a higher vertical resolution of the boundary layer."

Page 20, lines 19-20: In addition, the results have shown that a more detailed treatment of point source emissions including their vertical distribution, as well as the vertical model resolution itself, could become important when going to a horizontal model resolution of 1km.

Further changes

**Page 2, line 14-15:**
**The limit value was not exceeded at 3 stations instead of 1 station.**

"In Berlin, measured $NO_2$ annual means exceeded the European limit value of the annual mean at all but three measurement sites close to traffic in 2014 […]"

**Page 7,** line **2-4:**
**The sentence "All emissions from the energy industry..." is somewhat unclear and was changed to**

"In the TNO-MACC III inventory, all emissions from the energy industry within Berlin are point sources, and of the emissions from other industry sectors ca. 55% of the total emissions within Berlin for CO, 9-17% for particulate matter and up to 1% for other gases are included as point sources."

**Page 11, line 7-8:**
**There is a typo in this paragraph, it will be changed to**

"[…] while the bias of maximum temperatures modeled with 3km and 1km resolutions is mainly positive, the bias of the maximum temperatures modeled with a 15km resolution is negative."

**References**

The manuscript of Jänicke et al. (2016) has been accepted for publication.

The reference of Tuccella et al. (2012) had a typo and was corrected.

The references mentioned in the response to the referees were added to the bibliography.

---

## Author Comment (AC2) · 3 Nov 2016

**Authors' reply to comments of Referee 2**

We thank Referee 2 for the very helpful feedback on the manuscript. In the following, we address the comments, listing the Referee's comment (bold), our response (normal font) and changes in the manuscript, if applicable (indented). All references given in this response can be found in the bibliography of the discussion paper or below each response. In addition to the changes replying to the Referees' comments, we made few further minor changes to the manuscript, listed below.

**RC2-I**

**You used the RADM2 mechanism. This mechanism doesn't include several important biogenic VOCs, therefore it usually underestimates ozone compared to such gas chemistry mechanisms as RACM. I suggest this point to be discussed in the paper.**

RADM2 is a mechanism that is frequently used. We include a discussion of this as well as possible reasons for biases in ozone related to the chemical mechanism in the manuscript as follows.

> Page 4, line 29 – page 5, line 1: "The setup includes the RADM2 chemical mechanism with the Kinetic PreProcessor (KPP) and the MADE/SORGAM aerosol scheme. RADM2 has been used frequently in air quality applications over Europe (Mar et al., 2016, Im et al., 2015, Tuccella et al., 2014); the effect of this choice of chemical mechanism on modeled concentrations is further discussed in Section 4.2.1."

> Page 14, line 5-7: "This is consistent with what has been reported for a coarse European domain using RADM2 chemistry (Mar et al., 2016) and in line with previous studies showing a deficiency of many online-coupled models, including WRF-Chem with the RADM2 chemical mechanism, in simulating peak ozone concentrations (e.g. Im et al., 2015a). Mar et al. (2016) suggested that the low bias in modeled ozone could be partially explained by the inorganic rate coefficients used in the RADM2 mechanism."

**RC2 – II**

**You used 35 vertical levels, how thick is the first layer? You did simulations as high as 1km resolution, which certainly helps to capture spatial variability of the urban canopy and anthropogenic emissions in more detail. However, the vertical resolution remained the same. I think relatively coarse vertical resolution could explain some of the model deficiencies, especially the nighttime bias in the model.**

The first layer is at ca. 30m above the surface. We will add this information to the model description. Mar et al. (2016) have tested increasing the vertical model resolution and found that surface layer concentrations do not change much. For further discussion of the vertical model resolution as well as changes in the manuscript on the topic of vertical resolution please see our reply to the comments of Referee 1 ("general concern", "comment on page 10, lines 1-17", "summary").

> P. 4, line 28-29: "The model top is at 50 hPa, using 35 vertical levels. The first model layer is at approximately 30m above the surface, with 12 levels in the first 3km. "

**RC2 - III**

**Chapter 2.1- doesn't the ERA-INTERIM have 61 vertical levels?**

The full dataset has 60 vertical levels (http://www.ecmwf.int/en/what-vertical-resolution-data ). Here, we used the data interpolated to pressure levels. We made this clearer in the text as follows.

> Page 5, line 2-4: "We use the European Centre for Medium-Range Forecast (ECMWF) Interim reanalysis (ERA-Interim, Dee et al., 2011) with a horizontal resolution of 0.75°x0.75°, temporal resolution of 6h, interpolated to 37 pressure levels (with 29 levels below 50 hPa), as meteorological initial and lateral boundary conditions."

**RC2 - IV**

**2.2- WRF also uses more updated MODIS LU dataset. Here the USGS LU data is mentioned only.**

Thank you for pointing this out, we added it in the text.

Page 5, lines 9-10: "An analysis of the USGS land use data commonly used in WRF showed that the land cover of the region around Berlin is not represented well. In addition, the MODIS land use dataset as implemented in the WRF-Model from v3.6 only includes one category classifying urban areas. "

**RC2 – V**

**2.4- The stack height could be small, but the plume injection height due to buoyancy and momentum is higher. I think this could explain some of the NOx overestimations by the model at nighttime, when all the NOx from the point sources are emitted into shallow boundary layers.**

Thank you for your comment, we agree with you. We included this in the discussion of the original manuscript, e.g. page 19, lines 3-6: "As for the vertical distribution of emissions, Mar et al. (2016) state it has little impact on the model results. While this might hold for simulations of rural background air quality with domain resolutions of the order of 45km, the present results suggest that it is of higher relevance to distribute point source emissions into several vertical model levels when decreasing the model resolution and the resolution of the emission input data. " In addition, we elaborate further on this responding to the comments of Referee 1 (e.g. response to "RC1 - Summary").

**RC2 – VI**

**In WRF-Chem the vertical mixing of the chemical species are done somewhat differently than the meteorological fields. Did you consider testing sensitivity of the vertical mixing of the chemical species to various parameters/assumptions? The treatment of vertical mixing of the chemical species can explain some of the high NOx bias at night.**

We did not test the sensitivity of the vertical mixing of the chemical species to different parameters/assumptions for this study, as its main focus was to evaluate the model and test the sensitivity of the results to model resolution, resolution of the emission inventory as well as input parameters to the urban scheme and land use input data. However, we discuss the vertical mixing as a potential cause of the NOx bias at nighttime, e.g. page 13, lines 12-13: "The main reason for the nighttime overestimation is likely the model's underestimation of nighttime mixing […]". In addition, please also see our discussion in response to the comments of Referee 1 concerning the vertical resolution of the model (e.g. response to "General Concern", response to "RC1- Summary").

**RC2 – VII**

**In addition, I suggest showing vertical profiles or x-sections of the chemical species (e.g. NOx) to illustrate how deep the species were mixed in the model, especially during nighttime.**

Thank you for your suggestion. We have included exemplary vertical profiles in the supplement (see figures below). The figure shows mean NOx profiles in the lower troposphere simulated with a 1kmx1km model resolution at 00:00, 06:00, 12:00 and 18:00 UTC, for the base run (black) and S3 (downscaled emissions, red), for Amrumer Straße. Error bars give the $25^{th}$ and $75^{th}$ percentiles. We also included the following description in the manuscript:

Page 13, from line 12: "The main reason for the nighttime overestimation is likely the model's underestimation of nighttime mixing as discussed above. This is supported by the vertical distribution of NOx at several locations in the urban area, which shows a strong gradient between the first and second model layer (e.g. figure S10 in the supplementary material as an example)."

[Figure]

**RC2 - VIII**
**Can you show model-obs. comparisons similar to Figure 11 for NOy (if measurements are available) as well? The paper doesn't show any evaluations for CO, biogenic VOCs such as isoprene etc.**
Unfortunately, measurements of NOy are not available.  We do also not have observations of CO for urban background stations in Berlin for 2014. We agree with you that a comparison of modeled with observed CO might help to get additional insight into the cause of the NOx-bias and will consider this for further analyses. These are, however, beyond the scope of the present study. As for the comparison with VOCs, Churkina et al. (in preparation) compared modeled with observed isoprene and find that isoprene in the urban background is simulated reasonably well, but underestimated in urban parks and forests. The manuscript of Churkina et al. will include a detailed discussion of this comparison.

Further changes

**Page 2, line 14-15:**
**The limit value was not exceeded at 3 stations instead of 1 station.**
> "In Berlin, measured NO2 annual means exceeded the European limit value of the annual mean at all but three measurement sites close to traffic in 2014 […]"

**Page 7, line 2-4:**
**The sentence "All emissions from the energy industry..." is somewhat unclear and was changed to**
> "In the TNO-MACC III inventory, all emissions from the energy industry within Berlin are point sources, and of the emissions from other industry sectors  ca. 55% of the total emissions within Berlin for CO, 9-17% for particulate matter and up to 1% for other gases are included as point sources."

**Page 11, line 7-8:**
**There is a typo in this paragraph, it will be changed to**

"[…] while the bias of maximum temperatures modeled with 3km and 1km resolutions is mainly positive, the bias of the maximum temperatures modeled with a 15km resolution is negative."

**References**

The manuscript of Jänicke et al. (2016) has been accepted for publication.

The reference of Tuccella et al. (2012) had a typo and was corrected.

The references mentioned in the response to the referees were added to the bibliography.